# A single-cell atlas of conventional central chondrosarcoma reveals the role of endoplasmic reticulum stress in malignant transformation

Zezhuo Su [1,2,3,6], Joshua Wing Kei Ho [2,3,6], Raymond Ching Hing Yau[1], Ying Lee Lam[1], Tony Wai Hung Shek[4], Maximus Chun Fai Yeung[4], Hongtai Chen [1], Richard O. C. Oreffo[5], Kathryn Song Eng Cheah [2] & Kelvin Sin Chi Cheung [1✉]

The transformation of benign lesions to malignant tumours is a crucial aspect of understanding chondrosarcomas, which are malignant cartilage tumours that could develop from benign chondroid lesions. However, the process of malignant transformation for chondroid lesions remains poorly understood, and no reliable markers are available to aid clinical decision-making. To address this issue, we conducted a study analysing 11 primary cartilage tumours and controls using single-cell RNA sequencing. By creating a single-cell atlas, we were able to identify the role of endoplasmic reticulum (ER) stress in the malignant transformation of conventional central chondrosarcomas (CCCS). Our research revealed that lower levels of ER stress promote chondrosarcoma growth in a patient-derived xenograft mouse model, while intensive ER stress reduces primary chondrosarcoma cell viability. Furthermore, we discovered that the NF-κB pathway alleviates ER stress-induced apoptosis during chondrosarcoma progression. Our single-cell signatures and large public data support the use of key ER stress regulators, such as DNA Damage Inducible Transcript 3 (DDIT3; also known as CHOP), as malignant markers for overall patient survival. Ultimately, our study highlights the significant role that ER stress plays in the malignant transformation of cartilaginous tumours and provides a valuable resource for future diagnostic markers and therapeutic strategies.

[1] Department of Orthopaedics and Traumatology, School of Clinical Medicine, Li Ka Shing Faculty of Medicine, The University of Hong Kong, Pokfulam, Hong Kong SAR, China. [2] School of Biomedical Sciences, Li Ka Shing Faculty of Medicine, The University of Hong Kong, Pokfulam, Hong Kong SAR, China. [3] Laboratory of Data Discovery for Health Limited (D24H), Hong Kong Science Park, New Territories, Hong Kong SAR, China. [4] Department of Pathology, School of Clinical Medicine, Li Ka Shing Faculty of Medicine, The University of Hong Kong, Pokfulam, Hong Kong SAR, China. [5] Bone and Joint Research Group, Centre for Human Development, Stem Cells and Regeneration, Human Development and Health, Faculty of Medicine, University of Southampton, Southampton SO16 6HW, United Kingdom. [6] These authors contributed equally: Zezhuo Su, Joshua Wing Kei Ho. ✉email: kc81@hku.hk

Chondrosarcomas consist of a heterogeneous group of neoplasms that produce a cartilaginous matrix. After osteosarcomas, chondrosarcomas are the second most common primary malignancy of bone[1]. The vast majority of chondrosarcomas (85%) are conventional chondrosarcoma (CCS) and can be categorised according to their location in the affected bone; Namely, conventional central (75%), peripheral (10%), and periosteal (1%) chondrosarcomas which may develop from pre-existing benign cartilage tumour enchondromas, osteochondromas, and periosteal chondroma, respectively[2]. In addition to conventional chondrosarcoma, several rare subtypes of chondrosarcoma belonging to distinct grades have been identified, namely, dedifferentiated (high grade), mesenchymal (high grade) and clear cell (low grade) chondrosarcoma. Together, these rare subtypes represent only 10–15% of all chondrosarcomas[3]. The diagnosis of malignant transformation of benign tumours and the three-grade system for CCS is currently based on histological examination and is subjected to high interobserver variability, making clinical decision-making challenging[4]. The clinical management of benign cartilaginous tumours is usually conservative while chondrosarcoma requires surgical intervention. There is therefore an urgent unmet clinical need to identify markers which can predict clinical behaviour and guide clinical decision making. Furthermore, since chondrosarcoma are radioresistant and do not respond well to chemotherapy, a new therapeutic strategy is needed for the treatment of patients suffering from inoperable disease.

There are currently no large cohorts of systematic profiles tackling malignant transformation of benign cartilage tumours. Single immunohistochemical studies have suggested a few potential markers for diagnosis of malignant transformation but low sensitivity and specificity, limiting their clinical use. PTHLH expression was found to be retained while IHH was absent in enchondroma upon malignant transformation[5]. IHH expression was reported to be decreased in osteochondroma during malignant transformation when TGF-beta and BCL2 were increased[6,7]. Genetically, the frequent mutations such as *IDH1 and IDH2* observed in chondrosarcoma were also present in benign tumour and therefore provided limited information to guide clinical decision making[8,9]. Although defects in cell cycle regulators such as *TP53* and *CDKN2A* can induce the malignant transformation of a benign tumour in a mouse model[10,11], escape of cell cycle check point typically occurs during the progression from low to high-grade tumour in the patient[2,12–14]. Thus, the exact trigger leading to malignant transformation of benign cartilaginous tumours remains to be elucidated.

In the current study, the transcriptomic profiles from eight conventional central chondrosarcomas (CCCSs), one enchondroma, one chondroblastic osteosarcoma (COS), and one foetal femur were determined. This produced a single-cell atlas identifying eight cell clusters with distinct transcriptomes observed in CCCSs and the enchondroma. We found that the cellular response to ER stress plays a critical role in the malignant transformation in the conventional central chondrosarcoma variance. Furthermore, we conducted functional analysis using a patient-derived xenograft mouse model which showed induction of ER stress promote tumour growth while inhibiting ER stress suppressed tumour progression. Finally, we developed a prognostic model for cartilage tumours by applying our single cell signature to a large cohort of publicly available bulk RNA expression profiles of cartilage tumours. This model provided additional evidence that key ER stress regulators such CHOP could serve as an early marker. This study enhances our understanding of the molecular complexity of conventional central chondrosarcoma (CCCS) and offers valuable insight into the diagnosis and prognosis of CCCS as well as identifying therapeutic targets for the treatment of chondrosarcoma.

## Results

### Cellular heterogeneity of conventional central chondrosarcoma at single cell resolution.

All samples for single cell transcriptomic analysis were collected from patients who underwent surgery for chondroid lesions with no prior chemotherapy or radiotherapy with appropriate ethics approval (IRB: UW 16-2036). Seven samples were collected in the first batch: one benign enchondroma (Ben), two low-grade CCCS (Low_1 and Low_2), one medium-grade CCCS (Med) and two high-grade CCCS (High_1 and High_2). For comparison purposes, one COS sample was also included in our study as a point of reference for integrating single-cell data from different patients (Supplementary Fig. 1a, b). Histological analysis was conducted to ensure correct histological diagnosis of cartilage lesions (Supplementary Fig. 1c). Cartilaginous matrix was present in all samples as demonstrated by Safranin-O staining (Supplementary Fig. 1d). Tumours from benign, low- and medium-grade displayed abundance cartilaginous matrixes and low cellularity (collectively named differentiated tumours), while high-grade CCCS showed significantly higher cellularity and reduced matrix deposition. The COS showed atypical neoplastic chondrocytes with few neoplastic stromal cells as well as osteoid deposition. These observations confirmed the diagnosis of cartilage tumour in the COS sample obtained.

The seven primary tumour samples collected were analysed using single cell RNA sequencing (scRNA-seq) technology. For each tumour sample, the transcriptomic profile of approximately 4500 cells was generated, with a median of 2600 genes detected per cell (Supplementary Fig. 1a). The 25,739 cells in the single-cell atlas were separated into eight clusters and annotated based on differentially expressed genes (DEGs) and gene ontology (GO) term enrichment analysis (Fig. 1, Supplementary Fig. 2). The eight cell clusters were named as follows: Chon1, Chon2, High1, High2, Cos, Proliferative (Prol), Stromal (Stro) and Leucocytes (Leuk; Fig. 1a, b).

The Chon1 cluster expressed high levels of genes associated with cartilaginous matrix production; *COL21A*, *COMP* and *RARG* suggesting this cluster represents well differentiated neoplastic chondrocytes (Fig. 1c, d and Supplementary Fig. 2). The Chon2 cluster expressed lower levels of cartilaginous matrix genes and elevated levels of genes involved in the cellular response to ER stress namely, *DDIT3*, *HSPA5*, and *ATF5* (Fig. 1c, d and Supplementary Fig. 2). Critically, the benign sample contained mainly of cells from the Chon1 cluster, while low-grade and medium-grade samples contained Chon2 and Chon1 clusters. Together, these data suggest that Chon1 and Chon2 clusters represent early malignant transformation involving activation of the ER stress pathway and are discussed in detail below.

High-grade CCCS and COS samples each contained a cell cluster unique to the specific patient sample. High1 markers (e.g., *EIF3E*, *RPL30*, *RPL32*) enriched in ribosome biogenesis and translation GO terms (Fig. 1c, d and Supplementary Fig. 2). High2 expressed *TGFB1*, a key player in tumour invasion and metastasis[15,16], *ITGA10* a prognostic marker[17] and therapeutic target[18] for myxofibrosarcoma, and *EZR a* cell adhesion myosin regulating tumour proliferation and metastasis[19] (Fig. 1c, d and Supplementary Fig. 2). Furthermore, High2 markers were enriched in GO terms that are consistent with the invasive feature of the myxoid tumour[20]. For example, cell leading edge, ruffle, and lamellipodium, and actin fibre organisation (Supplementary Fig. 2). The major neoplastic cell population in High_1 and High_2 tumours can be further clustered into two subclusters (Supplementary Fig. 3). Differential expression genes (DEGs) enrichment analysis showed that High1a displayed a stronger chondrogenic characteristics and featured by oxidative phosphorylation while High1b was subjected to multiple cellular stresses

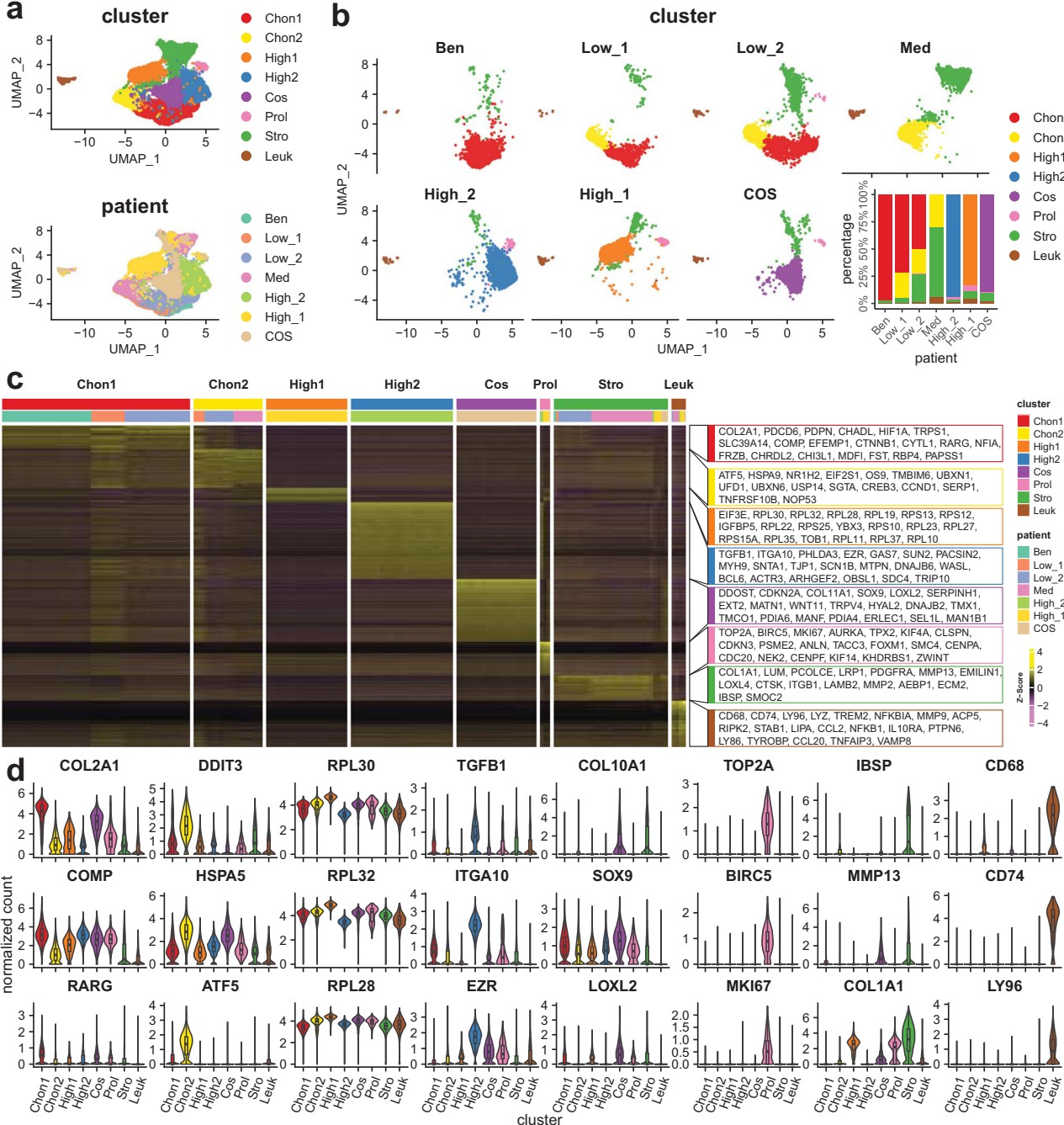

**Fig. 1 A single cell atlas of cartilage tumours. a** Representative UMAP plots illustrate the identification of cell clusters (top) and the origin of cell according to the patient sample (bottom). **b** The UMAP plot split by patient sample illustrates the cell clusters and the bar chart shows the abundance of clusters of each patient sample. **c** The heatmap shows the expression level of all specific marker genes in each cluster. **d** Violin plots display the expression of representative marker genes across cell clusters. The y axis shows the normalised read count. Box plots show the median, first and third quartiles, and minimum and maximum values within 1.5 times interquartile range. Chon1 Chondroid cluster 1, Chon2 Chondroid cluster 2, High1 major neoplastic cell cluster of High_1, High2 major neoplastic cell cluster of High_2, Cos major neoplastic cell cluster of COS, Prol Proliferating cluster, Stro Stromal cells except leucocytes, Leuk Leucocyte.

including hypoxia (Supplementary Fig. 3a, b). Similarly, High2a held a chondrogenic characteristics and characterised by synthesis of sterol and associated precursor cholesterol while High2b was characterised by response to steroid hormone, TGF-beta, NFkB, UPR, and p53 pathways, exocytosis, and secretion (Supplementary Fig. 3c, d). Chondroblastic osteosarcoma contains a unique cluster (Cos) expressing genes associated with hypertrophic chondrocyte differentiation; *COL10*, *SOX9*, and

*LOXL2* (Fig. 1c, d and Supplementary Fig. 2). This particular cluster, which is separate from the neoplastic chondrocyte clusters, typically associated with CCCS, was used as a point of reference for integrating the single-cell data.

In addition to the Chon1, Chon2, High1, High2 and Cos clusters described above, three cell clusters were found across the seven tumours and were annotated as Leucocytes (Leuk), Proliferative (Prol) and Stromal (Stro; Fig. 1a, b).

Prol contained markers enriched for cell division (e.g., *TOP2A*, *BIRC5*, and *MKI67*) as well as DNA repair (Fig. 1c, d and Supplementary Fig. 2). Prol was largely absent or in low abundance in Ben, Low and Med samples, and present in approximately 4% of the cell population in the two high-grade CCCS (High_1 and High_2) (Fig. 1b), indicating this cluster's association with an advanced tumour phenotype. Stro was characterised by markers (e.g., *COL1A1*/Collagen I, *LUM*, and *PCOLCE*; Fig. 1c, d) enriched in extracellular matrix organisation, ossification, biomineralization, and skeletal system development (Supplementary Fig. 2). Leuk had both myeloid lineage markers (e.g., *CD68*, *CD74*; and Supplementary Fig. 2) and lymphocyte lineage markers (e.g., *LY96*; and Supplementary Fig. 2), suggesting a heterogeneous cell population. Gene ontology enrichment analysis further supported the observation that an immune response was involved in both myeloid and lymphocyte (Supplementary Fig. 2).

Stro cell cluster was represented by seven subclusters with distinct markers: mineralisation stromal cell (miStro), ossification stromal cell (osStro), fibrocartilage chondrocyte (FC), cancer-associated fibroblast (CAF), granulating CAF (gCAF), vascular smooth muscle cell (vSMC), and endothelial cell (Supplementary Fig. 4a, b). miStro was almost exclusively present in Med (Supplementary Fig. 4a, b) and was characterised by co-expression of hypertrophic chondrocyte markers *COL10A1*/Collagen X, *COL1A1*/Collagen I, and bone mineralisation genes *IBSP* and *IFITM5* (Supplementary Fig. 5a–c). The expression of Collagen X and Collagen I in miStro was revalidated using immunohistochemistry (Supplementary Fig. 5c). The stromal cell population osStro was present primarily in benign and low-grade tumours (Ben, Low_1, and Low_2; Supplementary Fig. 4a, b). osStro cells expressed ossification and hypertrophic chondrocyte related genes *MMP13*, *KLF10*, *ENPP1*, as well as *COL1A1*/Collagen I (Supplementary Fig. 5a–c). Key gene ontology terms for miStro and osStro were bone mineralisation and ossification, respectively (Supplementary Fig. 5d). FC subcluster was identified by co-expression of *COL1A1*/Collagen I and gene markers (e.g., *COL2A1*/Collagen II, *COL9A1*, and *HAPLN1*) that enriched in chondrogenic characteristics (Supplementary Fig. 5). CAF was highly heterogeneous. Inflammatory CAF and antigen-presenting CAF related to immune response are reported in pancreatic ductal adenocarcinoma[21]. Here, we observed a gCAF subcluster (*CD68*, *HLA-DPB1*, and *CFD*; Supplementary Fig. 5a, b) in conventional central chondrosarcoma, which was characterised by granulocyte activation (Supplementary Fig. 5d). The other CAF (*COL1A1*/Collagen I, *COL3A1*, and *VCAN*; Supplementary Fig. 5a–c) was identified by extracellular matrix organisation and collagen fibril organisation (Supplementary Fig. 5d). In line with poor vascularisation of chondrosarcoma, two blood vessel cell populations were detected with low abundance (0.59% of total cells; Supplementary Fig. 4a, b). vSMC expressed *THY1*, *COL1A1*/Collagen I, *ACTA2*, and *RGS5* (Supplementary Fig. 5a–c). The other blood vessel cell type EC contained conventional markers (*PLVAP*, *CLDN5*, and *CD93*; Supplementary Fig. 5a, b)[22].

Leuk were further clustered into five subpopulations and annotated by conventional gene markers and ImmCluster[23]: M1 macrophage (M1), M2 macrophage (M2), osteoclast (Oc), T cell (Tc), and skeletal progenitor-like cell population (SP; Supplementary Fig. 4c, d). Monocyte lineage marker *CD68* was universally expressed in M1 macrophages, M2 macrophages, and osteoclasts (Supplementary Fig. 6a, b). M1 macrophages were characterised by genes associated with inflammatory response, including cytokine and chemokine such as *CLL3L1*, *CLL3*, and *CXCL2* (Supplementary Fig. 6b). In contrast, M2 macrophages (marked by *FCGR2B*, *FCGR3A*, *SLC2A1*, and *SPP1*) expressed a lower level of cytokine and chemokine and shown adaptation to a hypoxic environment (Supplementary Fig. 6). The other monocyte lineage derived cell population osteoclasts responsible for bone absorption contained distinct markers (*TNFRSF11A*, *NFATC1*, *ACP5*, and *SIGLEC15*; Supplementary Fig. 6b) and specific gene ontology terms (osteoclast differentiation, bone resorption, and bone remodelling; Supplementary Fig. 6c). Tumour infiltrated T cells expressed *TRAC*, *IL7R*, *CD8A*, and were negative for naïve T cell marker *CCR7* (Supplementary Fig. 6b). This cell population was observed to be highly proliferative (Supplementary Fig. 6d) Gene ontology terms (T cell activation, cytotoxicity, and CD8A-LCK complex; Supplementary Fig. 6c) indicated a $CD8^+$ cytotoxicity T cell population. Skeletal stem cells (SSCs) reside in the bone marrow stroma alongside hematopoietic stem cell[24] and play a supportive role for haematopoietic stem cell as well as cancer progression[25–27]. A small skeletal progenitor-like cell population (2–12%) was found in Leuk (Supplementary Fig. 4c, d). SP expressed human skeletal stem cell markers[28] ($PDPN^+$, $CD164^+$, and $NT5E^+$ and $MCAM^-$; Supplementary Fig. 6b). Moreover, SP markers enriched in gene ontology terms relevant to multipotency (Supplementary Fig. 6c).

Recent studies suggest that tumour associated stromal cells are highly heterogeneous. However, the extent of this heterogeneity within the chondrosarcoma microenvironment remains poorly characterised. By examining the Stro and Leuk populations, we observed that CCCS was supported by a small heterogeneous cell microenvironment (12 cell types accounting for 19.5% of total cells), maintained by a variety of stromal and leucocyte subpopulations (Supplementary Figs. 4–6). The stromal cells with mineralisation/ossification functions make up the majority of the stromal cells (45.8%; Supplementary Fig. 4b). Macrophages are the major infiltrated immune cells (75.7%; Supplementary Fig. 4d), consistent with literature[29].

**Copy number variation analysis reveals divergent progression of tumour development**. CNV is a common form of genetic alteration in cancers including chondrosarcoma[30] and is commonly used to identify malignant cells at single cell resolution[31]. We performed CNV analysis to examine the potential chromosomal alterations in the primary tumour samples collected. Genome-wide cell-type specific CNV patterns were inferred using Leukocyte subpopulation as the control (Supplementary Fig. 7a). The enchondroma sample (Ben) showed no significant CNV variation (Supplementary Fig. 7a) supporting the diagnosis of a benign tumour[32]. All CCCS (low, medium, and high-grade tumours) and the COS showed significant CNV changes in at least one cell cluster supporting the diagnosis of malignant tumours (Supplementary Fig. 7a). Notably, CNV levels of a region in chromosome 6 (grey boxes) was specifically lost in differentiated tumours except for the enchondroma (Low_1, Low_2, and Med), suggesting that this region may contain genes involved in early malignant transformation (Supplementary Data 1). High-grade chondrosarcomas as well as the chondroblastic osteosarcoma displayed a distinct CNV patterns (Supplementary Fig. 7a).

No evidence of significant CNV was found in Stro cluster of CCCS (also named as Stro1; Supplementary Fig. 7a) supporting the annotation of tumour associated stromal cell population, regardless of a small neoplastic stromal cell population (nStro) with osteoblastic characteristics presented in the COS (Supplementary Fig. 8). In contrast, large scale CNV were evidenced in the Chon1, Chon2, and Prol clusters of differentiated tumours except the enchondroma (Low_1, Low_2, and Med; Supplementary Fig. 7a). In addition to the common CNV region, distinct CNV regions were found for Chon1 and Chon2 in Low_2. The Chon2 cluster accumulated an additional copy number gain at

chromosome 19 (Supplementary Fig. 7a, b). Thirty-three genes, including *UBE2S* and *TRIM28*, showed significant copy number gain at this region (Supplementary Data 1). We found that the expression of *UBE2S* and *TRIM28* was correlated with the survival of the patient with chondrosarcoma (Supplementary Fig. 7d). UBE2S and TRIM28 interaction is reported to accelerate cell cycle[33], suggesting CNV at chromosome 19 exert significant impact on the progression of Low_2. In contrast, the Chon1 cluster revealed a loss in copy number at chromosome 11 (Supplementary Fig. 7a, c). This observation suggested a divergent progression of tumour development. Compared to Low_2, Low_1 was observed to be more homogenous with a similar CNV pattern between Chon1 and Chon2, indicating an earlier tumour development stage (Supplementary Fig. 7a).

Together our scRNA-seq analysis revealed a complex inter- and intratumor heterogeneity involving various cell types in chondrosarcoma including immune cells, stromal cells, and neoplastic chondrocytes. High-grade samples and COS each contained cell clusters unique to the patient sample collected. Critically, the Chon1 cluster with high expression of cartilaginous matrix associated genes was found in benign and low-grade tumour samples. In contrast, the Chon2 cluster was found in low-grade and medium grade tumour which could reflect a change in transcriptome during early malignant transformation from benign to malignant lesion.

### Comparing neoplastic chondrocytes with foetal femur chondrocytes.

Cartilage is typically found in joints, providing a smooth surface facilitating movement and serving as a shock absorber. Cartilage also plays a crucial role in endochondral ossification during skeletal development[34]. Chondrosarcomas commonly arise at the centre of bone termed CCCS[3]. It is thought that chondrosarcomas could arise in childhood during skeletal development[2]. To evaluate the differences between neoplastic chondrocytes found in cartilage tumours with normal chondrocytes at different stages of differentiation, we performed scRNA-seq analysis on a human foetal femur sample. The epiphysis and metaphysis of an 18-week gestation human foetal femur, comprised of chondrocytes from different stages of differentiation, were processed for scRNA-seq analysis. Employing standard bioinformatic analysis, six clusters were identified in the human foetal femur (Supplementary Fig. 9a). Resting chondrocytes (RC) were characterised by the expression of transcription factor *SOX9*, which is expressed during chondrogenic differentiation[35], as well as type II, IV, and VI collagens (*COL2A1*, *COL9A1*, and *COL11A1*) associated with the production of cartilage matrix Supplementary Fig. 9b. The overall gene expression profile of RC suggests these cells underwent active extracellular matrix organisation, collagen fibril organisation, and chondrocyte differentiation (Supplementary Fig. 9c). Hypertrophic chondrocytes (HC) were identified by the specific marker *COL10A1*, as well as *PTH1R* and *IHH* which have been described to regulate chondrocyte hypertrophic differentiation through a negative feedback loop (Supplementary Fig. 9b). The proliferating chondrocytes (PC) cluster expressed typical mitotic markers (e.g., *MKI67*, *TOP2A*, and *BIRC5*; Supplementary Fig. 9b and Supplementary Fig. 9c). An intermediate cluster between the RC and the HC clusters were found expressing genes *RPL5*, *NPM1*, *EEF1A1*, and *EIF3E* which are associated with ribosome assembly, rRNA processing, and translation and annotated as "Transitioning Cells"[36] (TC; Supplementary Fig. 9b and Supplementary Fig. 9c). Similar expression profiles were reported for differentiating cells during haematopoiesis[37] which underwent nucleolus assembly following mitosis[38]. In addition to the chondrocytes, we identified fibroblast-like (*COL1A1*, *VCAN*, and *VCAM1*) and endothelial

cells (*CLDN5*, *PLVAP*, and *CD93*; Supplementary Fig. 9b and Supplementary Fig. 9c).

Correlation of differential gene expression was used as a metric for identifying the similarity between neoplastic chondrocytes and foetal femur chondrocytes. Canonical correlation analysis (CCA) identified that Chon1 cells significantly resembled RC from the foetal femur (cosine similarity = 0.66). However, Chon2 cells did not correlate closely with chondrocytes at any differentiation stage examined (cosine similarity < 0.15; Supplementary Fig. 9d). Interestingly, no significant correlation was observed between neoplastic chondrocyte clusters and osteoarthritic chondrocytes clusters (Supplementary Data 2), suggesting that neoplastic chondrocytes resemble foetal chondrocytes more than osteoarthritic chondrocytes. We further examined the expression of Chon2 markers enriched in response to ER stress in foetal femur chondrocytes. Notably, genes associated with malignant transformation of differentiated tumours (*DDIT3*, *ATF5*, and *TRIB3*) were not detected during normal cartilage development (Supplementary Fig. 9e). These observations suggested that the Chon1 cluster, enriched in benign tumour and low-grade chondrosarcoma, retained a degree of RC characteristics while Chon2 cluster could be undergoing cellular reprogramming and malignant transformation. Furthermore, the ER stress response identified in cartilage neoplasm appeared to play a role in the pathological process of tumorigenesis but not in normal skeletal development.

### Cellular response to ER stress identified the malignant transformation of differentiated tumours.

The single-cell atlas and CNV inference demonstrated that the Chon2 cluster is a marker for the malignant transformation of differentiated tumours (consisting of Ben, Low, and Med). To identify the molecular signature of malignant transformation, we compared the Chon2 cluster with the Chon1 cluster. DEG analysis showed that 355 and 364 genes were upregulated in Chon2 and Chon1 clusters respectively (Supplementary Data 3). Notably, the Chon2 cluster markers were enriched in response to ER stress (e.g., *DDIT3*, *HSPA5*, and *TRIB3*), while gene markers of the Chon1 cluster were enriched for chondrocyte differentiation (e.g., *COL2A1*, *ACAN*, and *SOX9*; Fig. 2a). Protein expression of CHOP (encoded by *DDIT3*), HSPA5, and Collagen II (encoded by *COL2A1*) were revalidated by immunohistochemical analysis (Fig. 2b). Cellular response to ER stress is mediated by three cascades, namely, PERK, IRE1α, and ATF6[39]. PERK (*ATF4* and *NFE2L2*) and IRE1α (*XBP1* and *DNAJB9*) arms were found to be activated in Chon2 cells (Fig. 2c). Histologically, the high-grade chondrosarcomas demonstrated reduced cartilage matrix deposition compared with the low-grade and benign tumours. Further examination of Chon2 DEGs indicated genes also enriched in protein degradation associated pathways downstream of cellular response to ER stress, including unfolded protein response (UPR), proteasomal catabolism, and endoplasmic-reticulum-associated protein degradation (ERDA) pathways (Fig. 2d). These observations indicate that cellular response to ER stress may promote the progression of chondrosarcomas by reducing cartilage matrix production. In contrast, the ER stress evoked by the accumulation of Collagen X mutant in hypertrophic chondrocytes has been shown to cause Schmid metaphyseal chondrodysplasia (SMCD)[40]. The Chon2 cells shared 39 marker genes with SMCD chondrocytes in response to ER stress, including *DDIT3*, *HSPA5*, *TRIB3*, *SDF2L1*, and *ATF4* (Fig. 2e–g). Thus, in light of these observations, we propose that cellular response to ER stress is a molecular signature associated with the malignant transformation of differentiated tumours.

To validate the Chon2 cluster featured by cellular response to ER stress as a marker for CCCS, we further generated a new batch

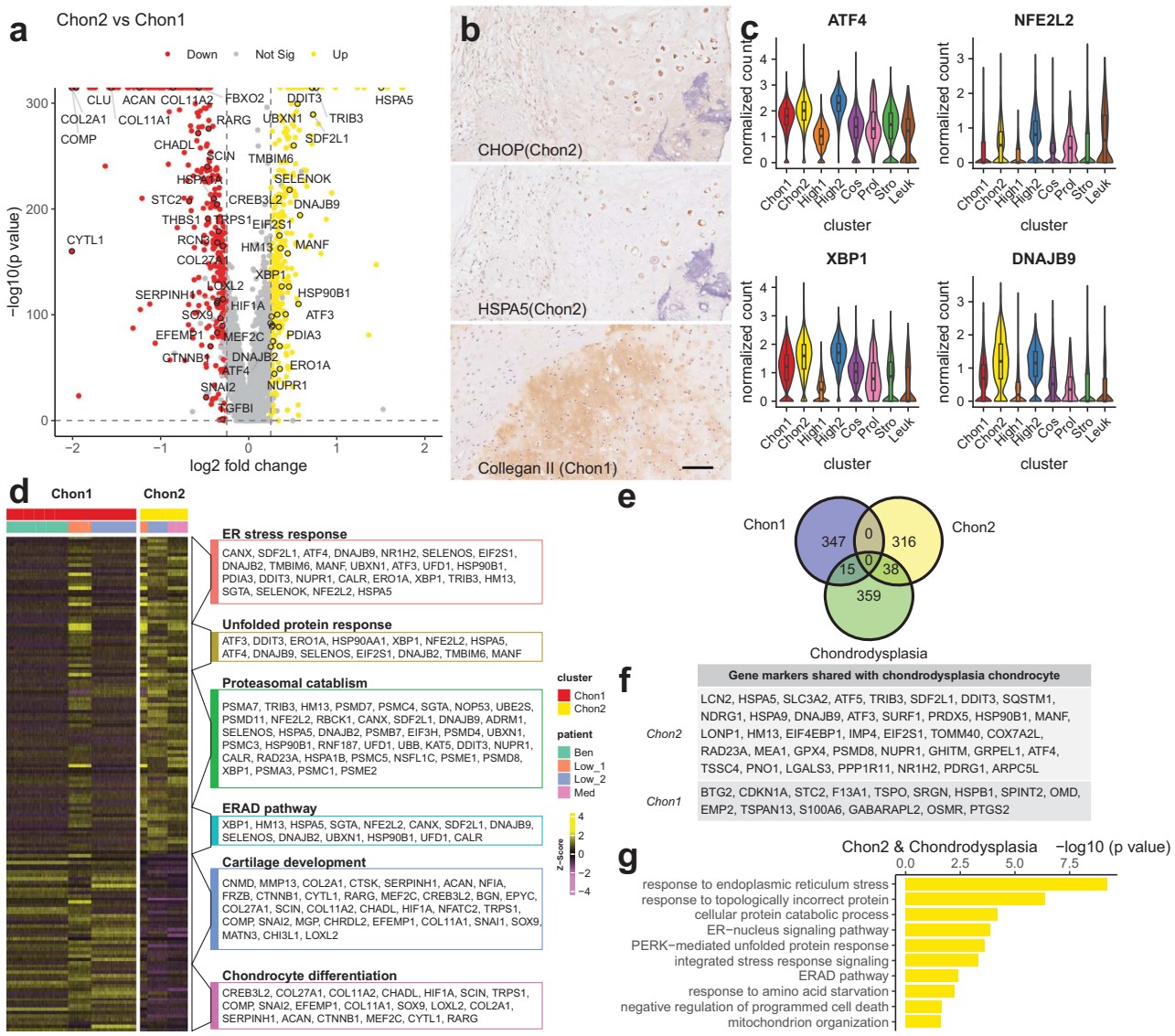

**Fig. 2 Response to ER stress indicates malignant transformation of cartilage tumours. a** Differential expressed genes (DEGs) analysis between Chon2 and Chon1. Genes enriched in chondrocyte differentiation (GO:0002062), response to endoplasmic reticulum stress (GO:0034976), and NIK/NF−kappaB signalling (GO:0038061) gene ontology terms are labelled. **b** Immunohistochemistry assay showing representative DEGs in situ. Scale bar = 100 μm. **c** Violin plots show specific genes regulating PERK (top) and IRE1α (bottom) arms of response to ER stress, respectively. Box plots show the median, first and third quartiles, and minimum and maximum values within 1.5 times interquartile range. **d** The heatmap displays DEGs enriched in representative biological processes. **e** Venn diagram shows the number of shared genes between chondrodysplasia chondrocytes gene markers in response to ER stress and differentially expressed genes of Chon2 and Chon1. **f** Shared genes list between chondrodysplasia chondrocytes and differentially expressed genes of Chon2 and Chon1 in Fig. 5e. **g** Representative enriched gene ontology terms of shared genes between chondrodysplasia chondrocytes and Chon2 cells. Box plots show the median, first and third quartiles, and minimum and maximum values within 1.5 times interquartile range.

of data from two low-grade CCCS and one medium-grade CCCS (*i.e.*, Low_L81, Low_L96, and Med_L80). Using standard bioinformatic pipeline, we characterised the cellular heterogeneity of the individual new samples (Supplementary Fig. 10a). In particular, we calculated the Chon2 and Chon1 scores for individual chondrocyte-like clusters. Analysis revealed that the Chon1 score (resembling chondrocyte characteristics) of at least one malignant cell population from one tumour (L81-1, L96-1, L96-4, and L80-1) was comparable to that of the reference (Chon1; Supplementary Fig. 10b), confirming the presence of a chondrogenic component and the diagnosis of chondrosarcoma. Critically, Chon2 score comparable to the reference (Chon2) was evident in two clusters (L81-2 and L96-2 from Low_L81 and Low_L96, respectively; Supplementary Fig. 10b). In addition,

DEGs analysis between L81-2 and L81-1 revealed that L82-2 expressed genes (e.g., *DDIT3*, *HSPA5*, and *TRIB3*) enriched in response to ER stress when L82-1 markers (e.g., *COL2A1*, *ACAN*, and *SOX9*) were enriched in chondrocyte differentiation (Supplementary Fig. 10c and Supplementary Data 4). Similarly, the expression signature of the L96-2 cluster also resembled the Chon2 cluster (Supplementary Data 4). These observations further supported the notion that cellular response to ER stress is a potential marker for CCCS.

**Cellular response to ER stress promotes the progression of chondrosarcoma in vivo.** CNV analysis suggested a divergent progression of differentiated tumours (Ben, Low_1, Low_2, Med).

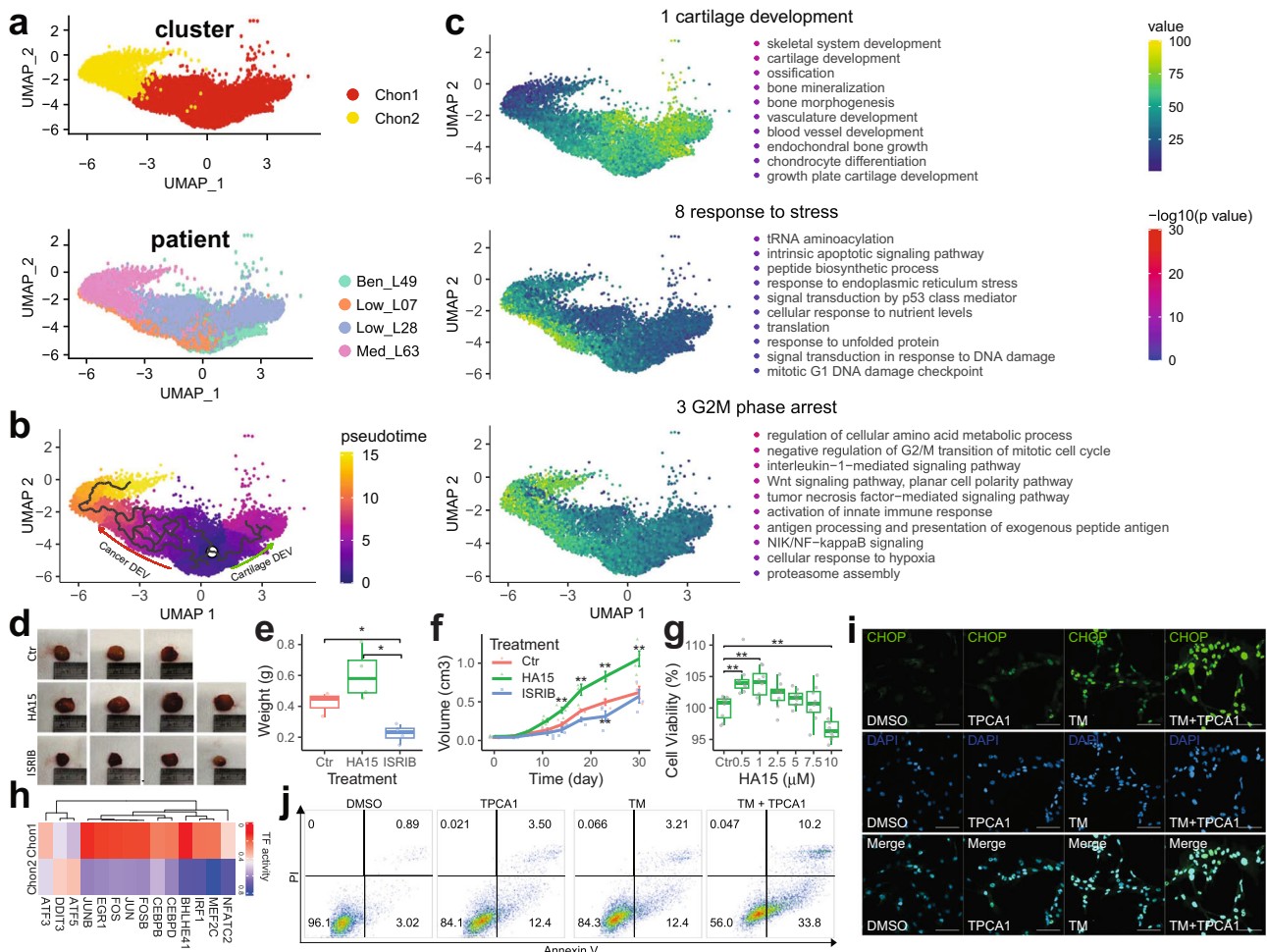

**Fig. 3 Discovering and validation of ER stress in tumorigenesis. a** UMAP plots show the major neoplastic cells from differentiated chondrosarcoma grouped by cluster (top) and patient (bottom), respective. **b** Pseudo-time trajectory of differentiated chondrosarcoma. The colour key from blue to yellow indicates the pseudo−time shaft. **c** UMAP plots showing the representative gene expression modules along the pseudo-time. Representative gene ontology and pathway terms for each module are displayed on the right. The Colour keys in the top and bottom indicate the average expression of genes scaled to range and -log10(p value), respectively. **d** Images of patient-derived xenografts of a high-grade conventional central chondrosarcoma after treatment with ER stress inducer (HA15), inhibitor (ISRIB), or Vehicle for one month in NOD SCID mice. **e** The box plot showing the tumour weight of xenografts in **d**. **f** The line plot shows the change of estimated tumour volume during treatment. Error bars represent standard deviation of the measurements. Error bars indicate the standard deviation of the measurements. A Student's t test was conducted to compare treatment groups to the control group. **g** Box plots illustrate cell viability under different ER stress pressure. Primary cells derived from the same patient were treated with 0.5, 1, 2.5, 5, 7.5, 10 μm HA15 or DMSO as a control for 48 h. Cells proliferation were evaluated using the Cell Counting Kit 8 (CCK-8). **h** The heatmap showing the active transcription factors of Chon1 and Chon2 clusters. Transcription factor is scored using AUCell. TF Transcription Factor. **i** Immunocytochemistry showing the localisation of CHOP (green) in SW1353. Cells were treated with NF-κB pathway inhibitor (TPCA1), ER stress inducer (TM), TPCA1 and TM, or DMSO as a control for 48 h. Scale bar = 100 μm. **j** The annexin V-FITC/PI flow cytometry assay shows the apoptosis levels of SW1353. Cells were treated with NF-κB pathway inhibitor (TPCA1), or DMSO as a control with or without ER stress inducer (TM) for 48 h. TM tunicamycin. Box plots show the median, first and third quartiles, and minimum and maximum values within 1.5 times interquartile range. P values by Student's T test are shown in the plot. *p value < 0.5. **p value < 0.01.

To investigate the molecular mechanism associated with tumour progression, bioinformatics inference of cellular trajectory of all cells in Chon1 and Chon2 using monocle 3 was performed. Since Chon1 cells resembled resting chondrocytes found within our normal human foetal femur sample, we used a resting chondrocyte gene marker *COL2A1* as a marker for cells of origin to define the root state (Fig. 3a, b; See Methods). This root-state was supported by the enrichment for benign tumour cells (Ben; Fig. 3a). Using the pseudo-time information, Monocle 3 identified ten co-expressed gene modules. In particular, we focused on three co-expressed gene modules: Module 1 (cartilage development, green arrow in Fig. 3b) represented benign development trajectory of chondrocyte (Fig. 3c); Modules 3 and 8 (cancer

development; red arrow in Fig. 3b) included genes that are associated with malignant transformation that involve early stress response including cellular response to ER stress (module 8) and later NF-κB pathway activation (module 3; Fig. 3c).

To confirm the role of ER stress in vivo, we established a patient-derived xenografts mouse model from a high-grade chondrosarcoma with IDH2 (H172R) mutation. In the study, primary tumour tissue was implanted subcutaneously into NOD SCID mice. One week after implantation, mice were treated with ER stress inducer, inhibitor, or vehicle for one month. The results showed that suppression of ER stress with an inhibitor ISRIB markedly attenuated the development of tumours, while the inducer HA15 promoted tumour growth (Fig. 3d–f). In contrast,

in vitro cell proliferation assay using primary cells from the same patient showed that low levels of ER stress increased cell proliferation while high levels of ER stress decreased cell proliferation (Fig. 3g). Studies have suggested that NF-κB is a survival pathway that suppresses CHOP-mediated apoptosis in response to ER stress in breast cancer[41]. Our trajectory analysis suggested that ER stress and apoptosis were early events during cancer development, followed by the activation of NF-κB pathway (Fig. 3c). In addition, SCENIC analysis identified DDIT3/CHOP as a key transcriptional regulator of Chon2 cell involvement in malignant transformation (Fig. 3h). Therefore, we hypothesise that NF-κB regulated the progression of differentiated tumours through alleviating CHOP-mediated apoptosis. Inhibiting NF-κB pathway induced CHOP expression and nuclei localisation (Fig. 3i), indicating NF-κB pathway regulated CHOP signalling in chondrosarcoma cells. Moreover, we mimicked ER stress response in Chon2 cells by applying tunicamycin (TM; an ER stress inducer) to chondrosarcomas cells. Upon TM treatment, CHOP was significantly upregulated and localised to nuclei (Fig. 3i). Under ER stress conditions, inhibiting NF-κB pathway not only further enhanced CHOP expression and nuclei localisation but also enlarged cell nuclei were observed (Fig. 3i), suggesting development of apoptosis. Therefore, we performed apoptosis analysis using flow cytometry. Notably, inhibiting NF-κB pathway significantly sensitised chondrosarcoma cells to ER stress induced apoptosis compared with DMSO control (Fig. 3j and Supplementary Fig. 11). Overall, these results indicated that ER stress and NF-κB signalling pathways played a critical role during the malignant transformation of chondrosarcoma.

**Prognostic model indicates a key ER stress regulator CHOP as a malignant transformation marker.** To further validate and support the cell type definitions found in our chondrosarcoma and benign enchondroma samples, we analysed a large publicly available bulk microarray dataset comprising 88 chondrogenic tumour samples[42]. The original study classified the patients based on two molecular tumour sub-types: E1 (chondrogenic subtype) and E2 (advanced subtype with poorer prognosis) based on gene expression. Using CIBERSORTx[43], we deconvoluted this micro-array gene expression data set using the seven cell types identified in our current single cell study. Critically, we observed a reduction of Chon1 cells and enrichment of Chon2 in E2 (poor prognosis group) compared to E1 (chondrogenic group; Fig. 4a), consistent with our observation that Chon2 is associated with malignant transformation of differentiated tumour. Moreover, Prol, which was mainly present in our high-grade tumours, was significantly enriched in the E2 subtype (Fig. 4a). Interestingly, High1 from a high-grade tumour was also enriched in E2 subtype (Fig. 4a). These observations demonstrated that our single-cell gene signatures were consistent with the large-scale bulk mRNA profiles.

Proliferation index is one of the critical factors in the grading of conventional chondrosarcoma[3]. In addition, the tumour microenvironment is typically correlated with the clinical outcomes of patients with chondrosarcoma[29,44]. We further hypothesised that the single-cell signature composition of proliferating cell population (i.e., Prol) and tumour stromal cells (i.e., Stro and Leuk) could prove useful for prognosis. To test this hypothesis, we used marker genes found in Stro, Leuk, and Prol clusters to characterise public bulk mRNA microarray profiles of clinical tumour samples consisting of benign tumour, CCS, and dedifferentiated chondrosarcoma[42]. Non-negative matrix factorisation (NMF) analysis stratified patients into five robust groups named according to their corelation with histological grades (by fisher's exact test; Fig. 4b and Supplementary Fig. 12a): Ben_bulk

($n = 15$), Med_bulk ($n = 18$), High_bulk ($n = 12$), Ded_bulk ($n = 8$), and Stro_bulk ($n = 21$; annotated by expression signature below).

The molecular features of distinct bulk groups offer new insights into the molecular complexity of chondrosarcoma across various subtypes and histological grades, as refined by their expression profiles. Leuk signature scores suggested that immune cells infiltration was a shared characteristic between Ben_bulk group and Ded_bulk group. Gene markers enrichment analysis further characterised the infiltrated immune cell population. Active immune response (*CD8A*, cytotoxicity T cell marker) and immunosuppression (*CD163*, M2 macrophage marker) were the characteristics of the infiltrated immune cells in Ben_bulk group and Ded_bulk group, respectively (Supplementary Fig. 12b, c). In line with high Prol signature scores, Ded_bulk group and High_bulk group expressed genes (e.g., *MKI67*) enriched in cell proliferation GO terms (Supplementary Fig. 12b, c). Besides, High_bulk group also expressed genes enriched in GO terms including response to hypoxia (*SLC2A1*, hypoxia marker), epithelial to mesenchymal transition (EMT; *SNAI2*, EMT regulatory transcriptional factor), and chondrocyte differentiation (*SOX9*; Supplementary Fig. 12b, c). The overall characteristics of Med_bulk group was chondrocyte development (Supplementary Fig. 12b, c). Stro_bulk group was overwhelming for bone signatures (*SP7*; bone development; Supplementary Fig. 12b, c). In addition, Stro_bulk was enriched for samples abundant with tumour stromal fraction (Supplementary Fig. 12d). These observations suggested that Stro_bulk may contain large amont of bony margin. Overall, the single-cell signature composition of Prol, Stro, and Leuk can classified chondrosarcoma into four distinct molecular groups associated with histological grades.

Notably, our algorithm stratified patients with low or medium-grade tumour into Ben_bulk or Med_bulk group for clinical decision making. We further characterised each bulk group in terms of signature scores of Prol, Stro, and Leuk. The Med_bulk group was negative for all three signatures (Fig. 4c). The Ben_bulk group and the High_bulk group were characterised by the signatures of Leuk and Prol, respectively (Fig. 4c). The Ded_bulk group had signatures of both Prol and Leuk. Stro_bulk group was identified by the signature of Stro (Fig. 4c).

To test the prognostic value of the bulk groups except Stro_bulk, we performed overall survival analysis. Patients in the Ben_bulk group, Med_bulk group, and High_bulk group displayed 100%, 76.9%, and 63.6% overall survival rates over five years, respectively (Fig. 4d). Therefore, our algorithm improved clinical diagnosis by stratifying patients with low or medium-grade tumours into two groups (Ben_bulk and Med_bulk) with more distinct clinical outcomes compared with histological grades (Supplementary Fig. 13a). On the other hand, the clinical outcomes for patients in the Ded_bulk group were the poorest, where overall survival rates dramatically dropped to 14.3% in less than two years (Fig. 4d). These survival curves demonstrated that the four-bulk-group system can predict clinical outcome of patients with chondroid lesions.

Our single-cell analysis suggested Chon2 as a cell cluster indicating malignant transformation. To confirm the Chon2 cluster signature could predict the malignant transformation in this large cohort, we evaluated Chon1 and Chon2 signature scores of individual bulk groups. The results showed that the Chon2 signature score was increased along with tumour grades while, the Chon1 signature score was decreased (Fig. 4e), supporting Chon2 cluster as a marker for malignant transformation. Next, we looked to identify gene markers for the diagnosis of malignant transformation. Examining genes involved in response to ER stress, we found that expression of *DDIT3/CHOP* as well as *HSPA5* was significantly increased in Med_bulk group compared

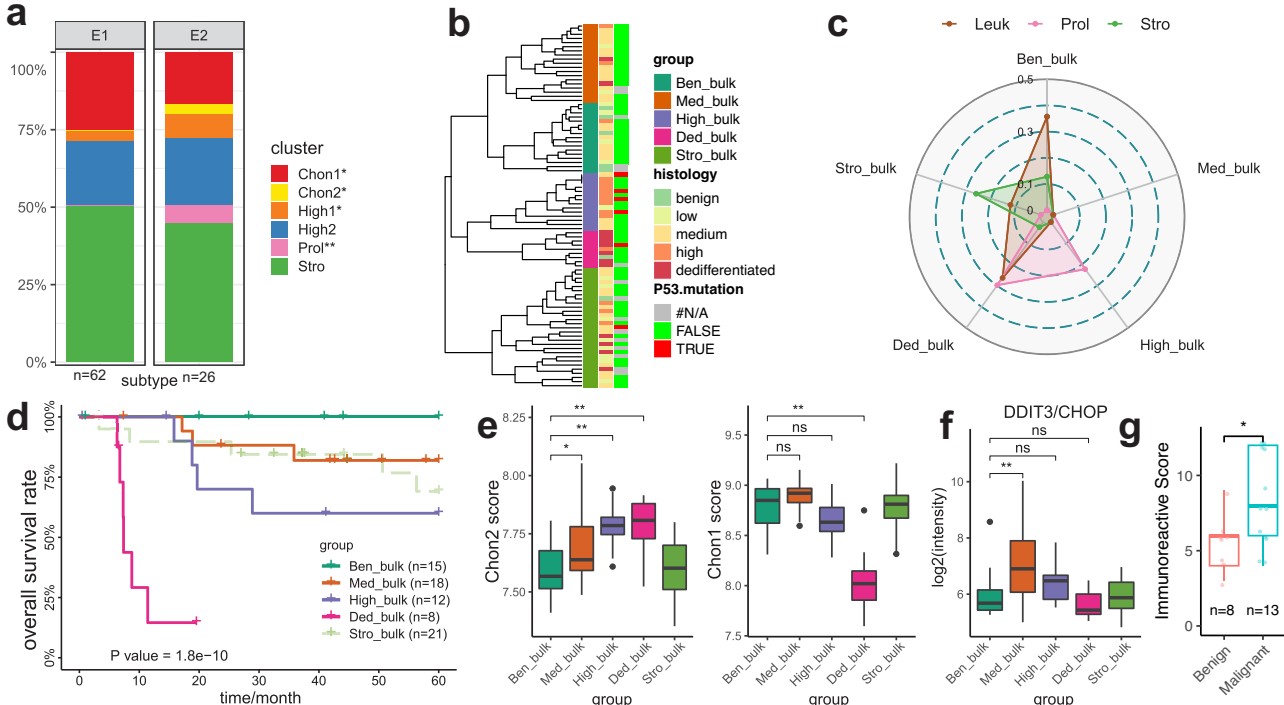

**Fig. 4 A prognostic model of chondrosarcoma indicates CHOP as a malignant transformation marker. a** Deconvolution of single-cell cluster abundance of 88 publicly available bulk mRNA profiles of benign lesions and conventional chondrosarcomas using CIBERSORTx. Bar charts showing average cluster abundance grouped by two expression subtypes (E1 and E2; defined in the original study). **b** Hierarchical clustering dendrogram demonstrates five bulk-mRNA-based groups. Non-negative matrix factorisation (NMF) analysis was performed on an expression matrix of Stro, Leuk, and Prol gene markers across 88 bulk mRNA profiles of benign lesions, conventional chondrosarcomas, and dedifferentiated chondrosarcomas. **c** The radar plot shows the signatures scores of Prol, Leuk, and Stro in each bulk group, respectively. Signature scores are defined by the canonical correlation coefficient between a single-cell cluster and a bulk expression group. **d** Overall survival rates of patients from individual bulk-mRNA-based groups estimated using Kaplan–Meier analysis. **e** Box plots show Chon1 and Chon2 cluster scores of bulk expression groups. The scores are evaluated by differentially expressed genes between Chon1 and Chon2 clusters. **f** The box plot displays the expression intensity of DDIT3/CHOP across bulk groups. **g** Immunohistochemistry assay shows immunoreactive scores of DDIT3/CHOP between eight benign and thirteen malignant tumours. The immunoreactive scores were quantified based on the proportion of positive cells and the intensity. Box plots show the median, first and third quartiles, and minimum and maximum values within 1.5 times interquartile range. P values by Student's T test are shown in the plot. ns not significant, *p value < 0.5. **p value < 0.01.

with Ben_bulk group (Fig. 4f and Supplementary Fig. 13b), suggesting an early marker for malignant transformation. To confirm the diagnostic value of CHOP, we analysed the survival data of the Ben_bulk and the Med_bulk groups involved in malignant transformation. Of note, all patients that died within five years expressed relatively high levels of *DDIT3/CHOP* or *HSPA5* (Supplementary Fig. 13c). We further confirmed the expression of *DDIT3*/CHOP using eight benign and thirteen malignant cartilage tumours samples. The immunoreactive score (IRS) of CHOP was higher in malignant cartilage tumours compared with benign lesions (Fig. 4g and Supplementary Fig. 13d). Overall, the prognostic model based on single-cell signature indicated *DDIT3*/CHOP as an early marker for chondrosarcoma.

### Discussion
The present study, to our knowledge, is the first single-cell transcriptomic analysis investigating cellular heterogeneity and malignant transformation in cartilage tumours. The single cell atlas reveals the role of endoplasmic reticulum stress in malignant transformation (Fig. 5a). In addition, an in vivo study showed that ER stress promoted the growth of CCCS in a patient-derived xenograft mouse model (Fig. 5b). Finally, a prognostic model supported key regulators of ER stress such as CHOP could serve as markers of CCCS (Fig. 5c). The present study identified ER stress as a marker for malignant transformation and we propose a

prognostic model for chondroid neoplasm to aid clinical decision making and which may inform the development of new treatment strategies for CCCS by targeting the ER stress pathway.

Benign and malignant tumours provide complementary perspectives on the mechanisms of cancer development[45]. However, tumour progression is rarely studied in benign tumours, although benign neoplasms may transform into malignant tumours as in the case of cartilage tumours. Clinical observations highlight the challenges faced for the histopathological diagnosis of malignant transformation in cartilage lesions compounding the clinical decision process[4,46,47]. Our single-cell analysis revealed ER stress regulators such as DDIT3/CHOP may be used to better differentiate benign chondroid lesion and early CCCS, providing an additional dimension for the diagnosis of malignant transformation in cartilage tumours. Although our analysis was based on pathological grading, subjected to interobserver variability, the diagnosis of the benign enchondroma was supported by the lack of significant CNV[30,48]. In addition to single-cell analysis from primary tumours, in vivo functional analysis demonstrated that ER stress promoted the progression of chondrosarcoma. Another piece of evidence that indicates the ER stress regulator CHOP as a malignant transformation marker derives from the prognostic model generated from a large cohort of patients with cartilage tumours. Finally, immunohistochemistry assay confirmed the upregulation of CHOP in malignant tumours compared to benign tumours.

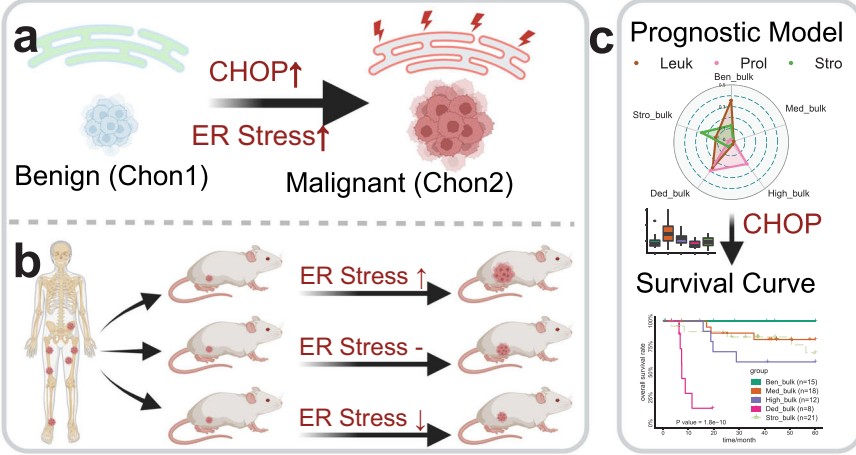

**Fig. 5 Summary of the study findings. a** Comparison between benign and malignant cells identifies ER stress response as a marker of chondrosarcoma. **b** ER stress induction promotes the growth of chondrosarcoma in a patient-derived xenograft mouse model. Similarly, inhibition of ER stress suppresses tumour growth. **c** A prognostic model developed with single-cell signatures further supports key ER stress regulators such as CHOP as malignant markers in a large cohort of patients with cartilage tumour. (Created with BioRender.com).

ER stress is a well-known hallmark of cancer; the evidence comes from the comparison between malignant and normal tissue regardless of benign lesion. The impacts of ER stress on cell fate depends on the extent and level of stress. Sustained ER stress has profound effects on immunosuppression, chemoresistance, angiogenesis, proliferation, and cancer cell survival[49–53]. Prolonged and intensive ER stress results in cell death via the CHOP-mediated apoptosis pathway[54]. ER stress induced the expression of SOX9 and FGF21 through ATF4/CHOP to reprogram differentiation of hypertrophic chondrocytes and active survival signalling pathway, respectively, in metaphyseal chondrodysplasia type Schmid[40,55]. ER stress in proliferating chondrocytes was reported to result in a longer growth plate and impair long bone growth[56]. Our analysis on primary tumour samples revealed the malignant transformation of benign cartilage tumours was associated with cellular response to ER stress. The chondrodysplasia mouse model showed that ER stress does not induce tumorigenesis. Therefore, ER stress may play an important role in the malignant transformation of benign tumours rather than tumour initiation. However, intensive ER stress reduces cell viability as demonstrated by NF-κB signalling pathway alleviated ER stress-induced apoptosis during the progression of chondrosarcoma.

Current diagnosis and clinical treatments for chondrosarcomas are primarily based on the classification of tumour grades[4]. Recently, two expression subtypes were suggested for the classification of cartilage tumours[42]. However, the two-subtype system does not fully represent the cartilage tumours of different histological subtypes and grades, therefore, providing limited information on the molecular complexity of chondrosarcoma. Based on the single-cell signatures of Prol, Stro, and Leuk, we propose a four-group system for the classification and prognosis of cartilage tumours. Proliferation index characterises advanced tumours (High_bulk and Ded_bulk groups). Active immune response can further characterise the low proliferative tumours into Ben_bulk group and Med_bulk group while immunosuppression characteristics distinguished the Ded_bulk group from the High_bulk group. The Stro_bulk is a distinct expression group that does not correlate with any histological group. The Stro_bulk expression signature is over-represented by tumour-associated stromal cells, indicating that Stro_bulk is likely a group of samples enriched for tumour stromal fraction and that the bulk expression profile of Stro_bulk does not reflect the signature of malignant cells. Taking advantage of scRNA-seq, we are able to overcome such limitation of bulk expression profile and characterise the molecular complexity of individual cartilage tumours.

Chondrosarcoma is the second most common primary tumour of the skeletal system. The major limitation of the current study is the limited sample size and while acknowledging the sample size was relatively small, the samples examined included a wide spectrum (from benign to high-grade) of samples. In the current study, we primarily focused on delineating the malignant transformation of the central variance of cartilage tumour to facilitate clinical decision making which could directly improve patient care. The malignant transformation of peripheral and periosteal tumours, which represents around 10 percent of cartilage tumours, remains to be further elucidated. The distinct characteristics of high-grade tumours could, to a degree, remain representative of patient-specific variances. Further investigations on a larger sample size of advanced chondrosarcoma may provide a comprehensive classification of advanced tumours and potential therapeutic target. Our study employed a high-grade chondrosarcoma-derived PDX mouse model to examine ER stress's impact on chondrosarcoma since PDX models derived from low and medium-grade tumours grow too slowly in vivo to investigate ER stress during chondrosarcoma progression. IDHs mutations are found in approximately 50% of chondrosarcomas, and mice with these mutations develop benign enchondroma-like lesions[57]. In contrast, mutant in a cartilage matrix protein, such as a 13-base pair deletion in Col10a1[55], elicits ER stress in chondrocytes in a mouse chondrodysplasia model. The tumour promoting role of ER stress during malignant transformation can be future explored using these two models. The proportion of stromal cells, including immune cells, in our study was insufficient for comprehensive analysis. However, our statistical model demonstrated these stromal cells' importance for patient stratification. Further investigation of the chondrosarcoma microenvironment could involve single-cell sequencing from enriched stromal cell populations in future studies.

## Methods

**Cartilage tumour samples.** The current study was approved by the Institutional Review Board of the University of Hong Kong/ Hospital Authority Hong Kong West Cluster (IRB reference number: UW 16-2036). Patient samples were obtained from the Queen Mary Hospital, Hong Kong, following patient consent.

Patient age, gender, anatomical site of surgical excision and clinical diagnosis were noted.

Fresh specimens were collected at the time of surgical resection or open biopsy. Diagnosis and grading were conducted by a group of pathologists and orthopaedic surgeons. Eight chondrosarcomas, one benign enchondroma, one chondroblastic osteosarcoma and one foetal femur was analysed. Specimens were transported on ice immediately following resection and kept at 4 °C within 24 h before processing.

**Tissue dissociation and cell purification**. Tumour tissues were washed 2-3 times vigorously with HBSS (Gibco, Cat. no. 14025076), then minced into small nodules and washed 2–3 times with HBSS again to prevent peripheral blood cell contamination. Tumour tissues were further minced into 1 mm³ nodules. We used a dissociation enzyme cocktail consisting of 0.4% (w/v) collagenase II (Gibco, Cat. no. 17101015), 0.4% (w/v) Hyaluronidase (Sigma, Cat. no. H3884-1G), 0.4% (w/v) dispase II (Sigma, Cat. no. D4693-1G) to digest the tissues. Tumour nodules were dissociated at 37 °C with shaking at 25 r.p.m for around 90 min. DNase I (Sigma, Cat. no. 4716728001) was introduced in the final 10 min to prevent cell aggregation. Cell suspensions were filtered using a 40 μm nylon cell strainer (Falcon, Cat. no. 352340). Residual red blood cells were kept as an indicator of peripheral blood cell contamination. Dissociated single cells were washed 2–3 times with HBSS containing 0.1% (w/v) BSA (Gibco, Cat. no. 15260073), then diluted to around 100 cells/μl for scRNA-seq.

**10x library preparation and sequencing**. Single-cell library and sequencing was performed at the Genomics and Bioinformatics Cores, Centre for PanorOmic Science, Faculty of Medicine, The University of Hong Kong. Briefly, single cell encapsulation and cDNA libraries were prepared using Chromium™ Single Cell 3' Reagent Kits v2 and Chromium™ Single Cell A Chip Kit. Cells were loaded according to standard protocol to capture 5000 to 10,000 cells/chip position per sample. All the remaining procedures, including the library construction, were performed according to the standard manufacturer's protocol. The library was then sequenced by Illumina NovaSeq 6000 using 150 nt paired-end sequencing with 100GB raw reads output per sample.

**Single-cell RNA sequencing data processing**. Reads were processed using Cell Ranger 3.0.0 pipeline with default settings. FASTQs generated from Illumina sequencing output were aligned to the human reference genome (GRCh38) using the STAR algorithm[58]. Next, we generated gene-barcode matrices for each sample by counting unique molecular identifiers (UMIs) and filtering non-cell associated barcodes. Finally, gene-barcode matrixes containing the barcoded cells and gene expression counts were generated.

The resulting gene-cell matrixes were subsequently imported into the Seurat (v3.1.5) R toolkit for further quality control and downstream analysis[59,60]. All functions were run with default settings, except those addressed below. Before sample integration, we inspected the samples individually and filtered cells with low quality (<200 genes/cell, <3 cells/gene, <5500 nFeature_RNA, and >5% mitochondrial genes percentage).

**Cell cycle phase, gene ontology term, and cluster scores analysis**. Cell cycle phase, GO ontology term, and cluster scores were estimated by a scoring strategy calculating the average expression of a gene list of interest[61], such as cell cycle phase marker genes, GO term associated genes, and cluster marker genes, which is implemented in Seurat as a function called AddModuleScore. For

the cell cycle phase, CellCycleScoring function derived from AddModuleScore function was employed. Briefly, S and G2/M stage scores were calculated by cell cycle phase specific gene lists. Cells showing anticorrelation with S and G2/M stage gene sets were annotated at G1 phase.

**Differential expression and functional enrichment analysis**. For single-cell RNA sequencing (scRNA-seq), FindAllMarkers function implemented in Seurat was applied on the count matrix (min.pc > 0.25, only.pos = T). Cluster marker genes were further filtered by pct.1 > pct.2, and p_val_adj < 0.05. Finally, marker genes shared by two or more clusters were removed. For differentially expressed genes between two single-cell clusters, FindMarkers function implemented in Seurat was applied on count matrix at RNA assay (min.pc > 0.25). Then, cluster marker genes were further filtered by p_val_adj < 0.05. For bulk mRNA profiles, group marker genes were calculated by limma package with default parameters[62]. Group marker genes were further filtered by adj.P.Val < 0.05 and logFC > 0.75. Functional enrichment analysis was performed using gprofiler2 R toolkit[63]. Functional terms with term size > 1000 were filtered.

**Batch effect correction and identification of distinct cell clusters**. An algorithm based on the detection of mutual nearest neighbours (MNNs) in the high-dimensional expression space implemented in SeuratWrappers was employed to remove batch effects and integrate samples from different patients[64]. Atypical neoplastic chondrocytes of chondroblastic osteosarcoma are a control that should not be clustered together with neoplastic chondrocytes of conventional central chondrosarcoma. After batch effect correction for a correction matrix with 50 principal components (PCs), we performed dimensionality reduction (Uniform Manifold Approximation and Projection, UMAP) and clustering of the top 40 PCs at a resolution of 0.2 using Seurat. Cluster scores were subsequently calculated as described above. Ambiguous cell subsets with cluster scores lower than 0.35 or differences with high scores greater than 0.35 within clusters were filtered, except for Stro and Leuk clusters which were filtered by total copy number variation (CNV; <30). Then, marker genes of distinct clusters were calculated.

For the foetal femur control sample, a standard analysis pipeline implemented as part of the software package Seurat was employed. We performed standard data processing (>200 genes/cell, >3 cells/gene, <5500 nFeature_RNA, and <5% mitochondrial genes percentage), normalisation, dimensionality reduction (top 20 PCs), clustering (resolution = 0.1), and differential gene expression analysis.

**Pseudo-time trajectory analysis**. The monocle3 (v0.2.1) was used to analyse single-cell trajectory of chondroid CCS to decipher the malignant transformation of benign tumour[65]. We used the integrated UMAP to calculate the trajectory graph. Root state was defined as a niche of 100 neighbouring cells with the highest mean expression level of COL2A1. In brief, top 100 cells ranked by COL2A1 expression levels were identified as the core of potential niches. Mean expression of COL2A1 of a niche was defined as the average expression of COL2A1 in the nearest 100 cells of corresponding core. Next, genes that change as a function along the progression were identified using graph_test function and filtered by q_value < 0.05 and morans_I > 1. Gene modules were clustered by find_gene_modules function, followed by gene ontology and pathway enrichment assay.

The velocyto (v0.17.17) was employed to estimate the spliced and unspliced reads[66]. Loom fields of individual patients were merged using SeuratWrappers (v0.2.0). Integrated UMAP was

directly fed to velocyto.R (v0.6) to project the RNA velocity vectors.

**Identification of active transcription factors**. We employed SCENIC (Single-Cell rEgulatory Network Inference and Clustering) R implement to infer gene regulatory networks and active transcription factors. SCENIC analysis was performed using default parameters as descript previously[67]. Two gene-motif rankings (500 bp upstream and 100 bp downstream of the transcription start site (TSS) or 10 kb around the TSS) were used to determine the search space around the TSS. The 20-thousand motif database was used for the inference of co-expression network (GENIE3) and the analysis of transcription factor binding motifs (RcisTarget).

**Large scale copy number variation estimation**. Large scale CNV was inferred using inferCNV R package by default settings[31]. To estimate the CNV landscape of malignant cells and to identify stromal cell populations, Leuk population with distinct leucocyte marker genes was used as normal CNV control to eliminate the somatic CNV. Notably, Leuk markers (only.pos = T, min.pct = 0.1, logfc.threshold = 0.1, p_val_adj < 0.05, and (pct.1 - pct.2) > 0) were filtered to reduce noise in leucocyte marker enriched regions such as HLA family enriched region in chromosome 6 and CC chemokine family enriched region in chromosome 17 (data not shown). Then total CNV levels of cells were calculated as the quadratic sum of CNV centered to 0[22]. The significant CNV genes at chromosome 19 of Low_2 is defined as genes located at the chromosome region where the distance of mean CNV between Chon2 cells and Chon1 cells is more than 0.1.

**Canonical correlation analysis**. Similarity between CCS single-cell clusters and the cell populations in foetal femur was measured by correlation of differential expression[68]. In brief, mean gene expression of a cluster or cell population was subtracted by the mean expression of all the other clusters or cell populations. Cosine similarity was then used to calculate the correlations between differential expression vectors between CCS and foetal femur as a metric for similarity.

**Identification of bulk mRNA profile groups using single-cell signatures**. Bulk mRNA microarray profiles of chondrosarcoma[42] were downloaded from EBML-EBI (E-MTAB-7264). Samples without histological grades were filtered. Following normalisation of raw probe set signal intensities using the robust multi-array average (RMA) algorithm, weighted correlation network analysis was conducted[69]. The 88 samples which passed the quality control were further analysed. The signature matrix of chondrosarcoma microenvironment and proliferation index was defined as the expression matrix of markers of Stro, Leuk, and Prol. Non-negative matrix factorisation was applied to the signature matrix with the number of rank set to four[70]. Five bulk groups were identified from the consensus matrix. Fisher's exact test was used to interrogate the correlation between expression groups and histological subtypes, and expression groups were annotated according to the most significant *p* value. Notably, the Stro_bulk group does not correlate with any histological group and is annotated by the expression signature overrepresented by bone-related features. Survival analysis on the bulk groups and histological grades was executed using the survival R package.

**Deconvolution analysis on bulk mRNA microarray profiles**. Dedifferentiated chondrosarcomas which had a non-cartilaginous component were filtered, since we were aiming for conventional central chondrosarcomas. CIBERSOERTx[43] was employed to estimate the abundances of clusters in bulk mRNA microarray profiles of chondrosarcoma samples which passed the quality control. Signature matrix was built using integrated data without Cos_L43 by default parameters except *q*-value < 0.05, 100 > No.Barcode Genes < 300, Min.Expression = 0.5, Replicates = 100, Sampling = 0.5. Cell fraction analysis was performed using S-mode batch correction and 500 permutations.

**Immunohistochemistry staining**. Primary tumour tissues were fixed in 4% paraformaldehyde at 4 °C for 48 h, followed by decalcification with EDTA buffer for one week at room temperature. Paraffin-embedding and sectioning were performed by following standard protocols. Sections of 5 μm thickness were melted, dewaxed, and rehydrated, followed by antigen retrieval using trypsin solution (Abcam, Cat. no. ab970). Blocking was performed at room temperature using 10% goat serum for 2 h, after which sections were incubated with primary antibodies at 4 °C overnight. Next, endogenous peroxidase was blocked by incubating for 10 min in 3% hydrogen peroxide solution. Following which sections were incubated with secondary antibodies (Dako, Cat. no. K5007-12) for 15 min and diaminobenzidine (Dako, Cat. no. K5007-12) for 3 min at room temperature. Finally, the sections were counterstained with haematoxylin, dehydrated, and mounted for whole-slide scanning.

Primary antibodies and their dilution rates are listed as follows: Collagen I (Abcam, Cat. no. ab34710, 1:2000), Collagen II (Abcam, Cat. no. ab185430, 1:2000), Collagen X (Abcam, Cat. no. ab58632, 1:2000), CHOP (Abcam, Cat. no. ab11419, 1:100), ATF5 (Abcam, Cat. no. ab184923, 1:100), MMP13 (Abcam, Cat. no. ab39012, 1:100), THY1 (Abcam, Cat. no. ab133350, 1:100), and CD68 (Abcam, Cat. no. ab955, 1:100).

**Immunocytochemistry staining**. SW1353 cells with 50% confluence were treated with 10 μM TPCA1 (Abcam, Cat. no. ab145522), 2 μM tunicamycin (SIGMA, Cat. no. T7765), 10 μM TPCA1 + 2 μM tunicamycin, or DMSO (SIGMA, Cat. no. D2650-100ML) in a chamber slide for 48 h. Then, cells were fixed with 4% PFA (SIGMA, Cat. no. 03112DH) for 15 min at room temperature followed by permeabilization with 0.5% Triton-x100 (SIGMA, Cat. no. SLBH9920V) for 5 min at room temperature. Next, blocking was performed at room temperature using 10% normal goat serum (Thermo, Cat. no. 50062Z) for 2 h, after which cells were incubated with the primary antibody CHOP (Abcam, Cat. no. ab11419; dilution, 1:100) at 4 °C overnight. Then, cells were incubated with fluorescence-conjugated secondary antibody Alexa Fluor® 488 (Abcam, Cat. no. ab150117; dilution. 1:1000) at room temperature for 1 h. Finally, the slides were mounted using a mounting medium with DAPI (VECTASHIELD, Cat. no. H-1200). Fluorescent images were obtained using a Carl Zeiss LSM980 laser scanning confocal microscope (Carl Zeiss, Germany) with a 20x objective.

**Animal study using a patient-derived-xenograft mouse cancer model**. The animal study is conducted under the protocol (CULATR-4997-18) approved by the Committee on the Use of Live Animals in Teaching and Research, The University of Hong Kong. Briefly, a patient-derived-xenograft nodule of 5 mm in diameter was implanted into an 8-week-old NOD SCID mouse subcutaneously. One week after implantation, mice were treated with ER stress inducer (HA15; 0.7 mg/mouse), inhibitor (ISRIB; 1 mg/mouse), or Vehicle intraperitoneally four times a week for one month. Mice were euthanised, and the tumours were resected for the measurement of tumour weight.

**Cell proliferation assay with cell counting Kit-8**. Primary tumour cells with 90% confluence were treated with HA15 or DMSO as a control in a 96-well plate for 48 h, followed by incubation with 10 µl of the CCK-8 solution in each well of the plate for 1 h. Finally, the absorbance at 450 nm was measured for the calculation of cell proliferation under different dosage of HA15.

**Flow cytometry analysis of apoptosis**. SW1353 cells with 90% confluence were treated with 10 µM TPCA1, 10 µM PRI-724, 2 µM tunicamycin, 10 µM TPCA1 + 2 µM tunicamycin, 10 µM PRI-724 + 2 µM tunicamycin, or DMSO in a six-well plate for 48 h. Flow cytometry analysis of apoptosis was performed following the manufacturer's protocol (BioLegend, Cat. no. 640928). In brief, cells were collected and resuspended in Annexin V Binding Buffer, followed by Pacific blue Annexin V and PI staining for 25 min at room temperature in the dark.

**Statistics and reproducibility**. All statistical analyses were conducted using R packages with specific packages employed for distinct analytical requirements. Continuous variables were expressed as mean ± standard deviation (SD). Differences between groups were assessed using Student's $t$ test for continuous variables, while Fisher's exact test was utilised for categorical variables. The range of sample sizes used was between three and 21 biological replicates. Canonical correlation analysis was performed to explore the relationships between two sets of variables and to identify the linear combinations that exhibited the highest correlations.

**Reporting summary**. Further information on research design is available in the Nature Portfolio Reporting Summary linked to this article.

## Data availability
All single-cell RNA sequencing raw and processed datasets generated in this study have been deposited at GSE184118. Gene markers and their associated enriched pathways can be accessed in Supplementary Data 3, 4, and 5. Source data for Figs. 1b (bar plot), d, 2c, g, 3e–g, and 4a, d–g are available in Supplementary Data 6.

## Code availability
Custom code used for this paper is available from GitHub at: https://github.com/zezhuo/Chon[71].

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

## Acknowledgements

We thank Danny Chan's group, Kathryn S. E. Cheah's group, William W. Lu's group, and Victor Y.L. Leung's group for their support and guidance in experiment. We would like to thank Prof. Oreffo at the Bone and Joint Research Group at University of Southampton for training our PhD students. Single-cell RNA sequencing, flow cytometry analysis, and imaging were performed at the Centre for PanorOmic Sciences, Li Ka Shing Faculty of Medicine, The University of Hong Kong. K.S.C.C. discloses support for the research of this work from the Research Grants Council [17126519] and the Health and Medical Research Fund [07182296]. J.W.K.H. discloses support for publication of this work from the Innovation and Technology Commission of Hong Kong [AIR@InnoHK].

## Author contributions

Z.S. carried out experiments, analysed data, and wrote the manuscript. J.W.K.H. supervised single-cell analysis and wrote the manuscript. R.C.H.Y. and Y.L.L. contributed to samples collection and clinical diagnosis. T.W.H.S. and M.C.F.Y. contributed to pathological diagnosis. H.C. contributed to the animal study. K.S.E.C. partially conceptualised the project. R.O. partially conceptualised the project and reviewed the manuscript. K.S.C.C. conceptualised the project, experiments design, supervised all aspect of the research process and manuscript writing.

## Competing interests

The authors declare no competing interests.
