## [Peer Review File · Communications Biology]

Reviewers' comments:

Reviewer #1 (Remarks to the Author):

The author examined the role of "A single cell atlas of chondrosarcoma reveals the role of endoplasmic reticulum stress in malignant transformation". The results indicated that endoplasmic reticulum (ER) stress is associated with malignant transformation between benign and malignant tumours. In addition, ER stress regulators such as the DDIT3; also known as CHOP as malignant markers in terms of overall survival. This study is well designed and provide new concept in this field.

Major points:

1. How to design and examine the CHOP effects in chondrosarcoma progression in further should be mentioned.
2. The cell model is important to examine the detail effect of DDIT3. The DDITs levels in normal chondrocyte and chondrosarcoma cell lines should be provided.
3. The biological process should be analysis to line DDITs with functional roles.
4. The limitation should be mentioned in Discussion.

Reviewer #2 (Remarks to the Author):

In this manuscript, the authors explored the cellular composition of chondrosarcoma at the single cell level and discovered a correlation between endoplasmic reticulum stress and malignant transformation. In addition, prognostic differences of chondrosarcoma have been revealed based on marker genes found in Stro, Leuk, and Prol clusters. More importantly, the authors have discovered a potential key gene CHOP involved in ER stress for diagnosis of malignant transformation. Overall, this work is interesting, but the following problems still exist:

Major point:

1. In single cell analysis, the marker genes used for cell cluster annotation need a functional description, especially for ER stress gene of Chon2 cluster. It's better to cite some references, such as 36271790, 31829268, 31377170.
2. How can ER stress be associated with malignant transformation based solely on-stage information from samples? It is best to give a direct explanation from the functional level of the marker genes.
3. To evaluate the differences between neoplastic chondrocytes found in cartilage tumours with normal chondrocytes, authors used fetal femur chondrocytes which may be composed of unique cellular component. The authors should compare neoplastic chondrocytes with adult adjacent normal samples or samples from healthy donors.
4. As chondrosarcoma at different locations have unique genetic mutations, the authors can further explore the association between central chondrosarcoma specific Chon2 cluster and genetic variation.
5. Among the known subtypes, authors observed enrichment of Chon2 and High1 in E2 (advanced subtype with poorer prognosis). Why did authors not first use the markers of these two clusters for prognostic analysis, but instead of Prol, Stro and Leuk?
6. The authors should further explore the impact of ER stress on the prognosis of chondrosarcoma and its mechanism, and investigate the relationship between ER pressure and proliferation.

Minor point:

1. The patients of five groups stratified by NMF were named according to the correlation with histological grades. It is better to describe the naming principles in detail.
2. In the prognosis analysis of the five groups, authors need to describe the Stro_bulk group more

Reviewer #3 (Remarks to the Author):

The authors of this manuscript generated and analysed single cell RNA sequencing data of one enchondroma, 5 chondrosarcomas and one chondroblastic osteosarcoma. Clustering analysis revealed multiple clusters corresponding to low/medium chondrosarcoma (chon1/chon2), stroma and leukocyte clusters, but also patient specific clusters. Deeper analysis of the chon1 and chon2 clusters revealed that these could represent early malignant transformation and DEG analysis between the two groups showed genes involving the activation of the ER stress pathway upregulated in chon2 cluster. To confirm that ER stress pathway is associated with malignant transformation an in vivo model was used where inhibition of ER stress reduced tumor weight. CNV analysis showed large scale CNVs in all chondrosarcoma samples, also displaying some heterogeneity. Single cell RNAseq data was validated using a new batch of chondrosarcoma patients and similar malignant and stromal cell populations were found.

While a large amount of work was performed, the comparisons made are debatable and some conclusions are not fully supported by the data shown. More specifically:

- Chondrosarcomas are a heterogeneous group of tumors and in the main single cell analysis only 5 chondrosarcomas of 3 different grades are included. There are patient specific clusters which already suggests that the chondrosarcomas are heterogeneous between patients, which makes it not fair to compare them. The patient specific clusters are disregarded (which is a pity) and most of the further analyses are based on cluster chon1/chon2, which only consists of 1 enchondroma and 3 chondrosarcomas. Furthermore, by looking at the clusterings in figure 1A and 1B, there is still some patient specificity. The data analysis would improve and be more convincing if more samples can be added.
- The inclusion of a chondroblastic osteosarcoma is not well explained and it can be discussed whether it makes sense since it is a different sarcoma type. It can be seen in figure 1A that it clusters separate. The advice would be to remove that sample
- Immune cells are found in the scRNA-seq analysis, it is a pity that a more in depth analysis of the immune microenvironment is lacking, which would have improved the paper
- The CNV analysis per tumor could be extended. It would be nice to see the copy number variation per cell and see how heterogeneous they are within a patient. Comparison between the different samples is less informative as CNVs are known to be highly heterogeneous between samples.
- It is tested whether the cellular response to ER stress is specific to central chondrosarcomas (line 280). In order to test this only one peripheral chondrosarcoma sample was analyzed. A larger panel of tumors should be analyzed before conclusions can be drawn. Here the inclusion of an unrelated tumor such as osteosarcoma as a control can be considered.
- Clinical information and IDH /EXT status of the samples should be provided
- Reference 25 does not seem to be right? Reference 25 contain bulk RNA samples from hepatocellular carcinomas, not chondrogenic tumor samples?
- Reference 3: needs update to 2020 WHO classification 5th edition
- It is not clear how the cell populations described in lines 285-289 were established and what the exact characteristics are.
- It is not clear what kind of primary tumour tissue was implanted subcutaneously to NOD SCID mice (line 321)
- The survival curves shown merely represent the diagnosis / tumor type than a gene expression cluster: this was not shown to be independent and therefore these curves are misleading and should be removed
- The immunohistochemistry shown only confirms the expression, not the differential expression between groups, and is therefore not informative. Moreover, the DDIT3 staining does not seem specific as it should be nuclear staining only.

Response to referees

Reviewer 1:

Comment 1: How to design and examine the CHOP effects in chondrosarcoma progression in further should be mentioned.

Response:

IDHs mutations are found in approximately 50% of chondrosarcomas, and mice with these mutations develop benign enchondroma-like lesions rather than malignant phenotype⁴³. On the other hand, mutant in a cartilage matrix protein, such as a 13-base pair deletion in Col10a1⁴¹, elicits ER stress in chondrocytes in a mouse chondrodysplasia model. The tumour promoting role of ER stress during malignant transformation can be future explored using these models with CHOP deletion.

Corresponding updates: Line 541-546 (All lines are numbered in the "All Markup" view)

Comment 2: The cell model is important to examine the detail effect of DDIT3. The DDIT3 levels in normal chondrocyte and chondrosarcoma cell lines should be provided.

Response:

We conducted additional experiments using the SW-1353 cell line derived from a medium-grade chondrosarcoma. A low level of DDIT3 was detected in this cell line without a clear nuclear signal (Fig. 3i). As low- and medium-grade chondrosarcomas grow slowly, developing a highly proliferative cell line might alter the primary tumour's properties, including the cellular response to ER stress. In our experiments, we treated the SW-1353 cell line with tunicamycin to mimic the ER stress response in primary tumours.

Foetal femur contains chondrocytes at different developmental stages and serves as an appropriate cell phenotype to compare chondrosarcoma samples. DDIT3 was not found to be expressed in foetal chondrocytes, suggest the presence of DDIT3 expression is part of a pathological process involved in the development of chondrosarcoma (Supplementary Fig. 5d). Besides, Matt et al. demonstrated that DDIT3 is minimally detected in normal human chondrocytes but significantly increases in advanced osteoarthritic chondrocytes (PMID: 24351550, Fig. 1A).

Corresponding updates: Line 369-376; Figure 3i

Comment 3: The biological process should be analysis to line DDIT3 with functional roles.

Response:

In this study we showed that chondrosarcomas have high level of DDIT3 compared to benign lesions, and DDIT3 is not expressed in normal chondrocyte development. Studies have suggested that DDIT3 mediates cell apoptosis under high levels of ER stress. Our additional analysis indicated that during chondrosarcoma progression, ER stress signalling including DDIT3 was alleviated while the NF- κ B signalling pathway increased, suggesting a critical role for the NF- κ B signalling pathway in controlling ER stress levels and extent (Fig. 3c). However, the tumour promoting role of ER stress/DDIT3 during malignant transformation need to be future explored.

Corresponding updates: Line 342-347, 360-379; Figure 3c-j

Comment 4: The limitation should be mentioned in Discussion.

Response:

Our study used a high-grade chondrosarcoma-derived PDX mouse model to examine ER stress's impact on chondrosarcoma since PDX models derived from low- and medium-grade tumours grow too slowly to investigate ER stress during chondrosarcoma progression in vivo. IDHs mutations are found in approximately 50% of chondrosarcomas, and mice with these mutations develop benign enchondroma-like lesions⁴³. On the other hand, mutant in a cartilage matrix protein, such as a 13-base pair deletion in Col10a1⁴¹, elicits ER stress in chondrocytes. The tumour promoting role of ER stress during malignant transformation can be future explored using these models.

Chondrosarcoma, like cartilage tissue, is known for poor vascularization. The proportion of stromal cells, including immune cells, in our study was insufficient for comprehensive analysis. However, our statistical model demonstrated these stromal cells' importance for patient stratification. Further investigation of the chondrosarcoma microenvironment could involve single-cell sequencing from enriched stromal cell populations in future studies.

Corresponding updates: Line 528-554

Reviewer 2:

Major point:

Comment 1: In single cell analysis, the marker genes used for cell cluster annotation need a functional description, especially for ER stress gene of Chon2 cluster. It's better to cite some references, such as 36271790, 31829268, 31377170.

Response:

We add supplementary data in a separated file to further characterized malignant cells and tumour associated stromal cell. For the stromal cells including immune cells that are commonly present in all kinds of tumours, we used conventional gene markers and packages (PMID: 36271790) for annotation.

For the malignant cells, two subtype was proposed by Rémy et al. (PMID: 31604924) based on expression profile of 102 cartilage tumours however, due to the limitation of experimental tools, the molecular subtype of chondrosarcoma is poorly understood. We present the first single-cell analysis study of chondrosarcoma. The malignant cell clusters were characterized based on conventional gene markers and gene set enrichment analysis. In addition, we compared neoplastic chondrocyte populations with developing chondrocytes from foetal femur and osteoarthritic chondrocytes. We found that neoplastic chondrocytes resemble a specific cell population found in foetal derived chondrocytes more than any osteoarthritic chondrocytes (Supplementary Fig. 5c and Supplementary Table 4).

Corresponding updates: Supplementary data - "*Characterization of Prol, Stro, and Leuk cell clusters*"; Line 251-254; Supplementary Table 1.9-1.10, 2.8-2.9; Supplementary Table 4

Comment 2: How can ER stress be associated with malignant transformation based solely on-stage information from samples? It is best to give a direct explanation from the functional level of the marker genes.

Response:

ER stress is a well know marker of cancer; evidence comes from the comparison between normal and tumour tissues. Sustained ER stress has profound effects on immunosuppression, chemoresistance, angiogenesis, proliferation, and cancer cell survival³⁵⁻³⁹. Prolonged and intensive ER stress results in cell death via the CHOP-mediated apoptosis pathway⁴⁰. However, the role of ER stress during malignant transformation is not fully understood. We propose a potential role of ER stress during malignant transformation, considering the notion that the impact of ER stress on cell fate depends on the level and extent of the stress. In our study, we found that increased ER stress is associated with the unfolded protein response, which may decrease cartilage matrix protein production and increases cellularity for progression and metastasis (Fig. 2d). *In vivo* PDX mouse model has demonstrated ER stress promote the growth of chondrosarcoma. Additionally, we performed further analysis and *in vitro* experiments, showing that the NF- κ B signalling pathway may promote chondrosarcoma progression by alleviating ER stress-induced apoptosis (Fig. 3g-j).

Corresponding updates: Line 342-347, 360-379; Figure 3c-j

Comment 3: To evaluate the differences between neoplastic chondrocytes found in cartilage tumours with normal chondrocytes, authors used fetal femur chondrocytes which may be composed of unique cellular component. The authors should compare neoplastic chondrocytes with adult adjacent normal samples or samples from healthy donors.

Response:

We agree that comparing neoplastic chondrocytes with adult adjacent normal samples or samples from healthy donors would be beneficial. However, as conventional chondrosarcomas commonly arise at the centre of bone rather than the joint cartilage, the adjacent normal tissue is bony tissue which consist of trabecular bone and bone marrow cells.

Given chondrosarcoma consists of growing chondrocytes, we first compared neoplastic chondrocytes with chondrocytes of different developmental stages from foetal femur. In response to this comment, we have further compared neoplastic chondrocytes with publicly available osteoarthritic chondrocytes. We found that neoplastic chondrocytes resemble foetal chondrocytes more than osteoarthritic chondrocytes (Supplementary Fig. 5c and Supplementary Table 4).

Corresponding updates: Line 251-254; Supplementary Table 4

Comment 4: As chondrosarcoma at different locations have unique genetic mutations, the authors can further explore the association between central chondrosarcoma specific Chon2 cluster and genetic variation.

Response:

In response to the suggestion to explore the association between the Chon2 cluster and genetic variation in conventional chondrosarcoma, we have further investigated genes located in the copy number gain region specific to the Chon2 cluster.

Thirty-three genes, including *UBE2S* or *TRIM28*, shown significant copy number gain at this region (Supplementary Table 3.2). The expression of *UBE2S* and *TRIM28* was correlated with the survival of patient with chondrosarcoma (Supplementary Fig. 4d). Moreover, *UBE2S* and *TRIM28* interaction is reported to accelerate cell cycle¹⁸, suggesting CNV at chromosome 19 exert significant impact on the progression of Low_2.

Corresponding updates: Line 196-202; Supplementary Table 3.2; Supplementary Fig. 4d

Comment 5: Among the known subtypes, authors observed enrichment of Chon2 and High1 in E2 (advanced subtype with poorer prognosis). Why did authors not first use the markers of these two clusters for prognostic analysis, but instead of Prol, Stro and Leuk?

Response:

We initially attempted to use the signature of all neoplastic cell clusters for prognostic analysis, but the results did not represent significant biological meaning. The molecular signatures of high-grade tumours identified in our study are diverse, as neoplastic cells of individual high-grade tumours form unique cell clusters as well as potential patient sample variation. In contrast, signatures of Prol, Stro, and Leuk clusters are common across patients and stratified patients into five groups with distinct molecular signatures based on bulk expression profiles.

Comment 6. The authors should further explore the impact of ER stress on the prognosis of chondrosarcoma and its mechanism, and investigate the relationship between ER pressure and proliferation.

Response:

Additional experiments showed that low levels of ER stress promote the proliferation of primary chondrosarcoma cells while high levels of ER stress inhibit proliferation (Fig. 3g). Low level of ER stress induced promote chondrosarcoma cells growth while high level of stress inhibitor cells growth.

Corresponding updates: Line 360-363; Figure 3g

Minor point:

Comment 1: The patients of five groups stratified by NMF were named according to the correlation with histological grades. It is better to describe the naming principles in detail.

Response:

We apologize for any confusion regarding the naming principles for the five patient groups stratified by NMF. We used Fisher's exact test to interrogate the correlation between expression groups and histological subtypes, and expression groups were annotated according to the most significant p value. Notably, the Stro_bulk group does not correlate with any histological group and is annotated by the expression signature overrepresented by bone-related features.

Corresponding updates: Line 709-713

Comment 2: In the prognosis analysis of the five groups, authors need to describe the Stro_bulk group more.

Response:

We acknowledge the need to provide more information on the Stro_bulk group in our prognosis analysis. Stro_bulk is a distinct expression group that does not correlate with any histological group. Its expression signature is overrepresented by bone-related features. Additional analysis further confirmed that the top signature genes of Stro_bulk is highly expressed in a tumour-associated stromal cell population (Supplementary Fig.8d). Therefore, Stro_bulk is likely a group of samples enriched for tumour stromal fraction, and the bulk expression profile of Stro_bulk does not reflect the signature of malignant cells. Taking advantage of scRNA-seq, we are able to overcome such limitation of bulk expression profile and characterize the molecular complexity of individual cartilage tumours better.

Corresponding updates: Line 421-424; Supplementary Fig.8d

Reviewer 3:

Comment 1: Chondrosarcomas are a heterogeneous group of tumors and in the main single cell analysis only 5 chondrosarcomas of 3 different grades are included. There are patient specific clusters which already suggests that the chondrosarcomas are heterogeneous between patients, which makes it not fair to compare them. The patient specific clusters are disregarded (which is a pity) and most of the further analyses are based on cluster chon1/chon2, which only consists of 1 enchondroma and 3 chondrosarcomas. Furthermore, by looking at the clusterings in figure 1A and 1B, there is still some patient specificity. The data analysis would improve and be more convincing if more samples can be added.

Response:

We appreciate the concern regarding the small sample size and inter-tumoral heterogeneity in our analysis. Our primary focus is on the malignant transformation between benign and low-grade tumours. We have narrowed down our analysis to 4 (out of 6; 66.7%) cartilage tumours associated with malignant transformation and identified ER stress as a key factor involved in malignant transformation. This finding is further supported by animal studies and a large cohort of bulk expression profile of chondrosarcoma. In addition, we added a new data from one patient with conventional chondrosarcoma and a new batch of immunohistochemistry staining to support ER stress/CHOP is associated with malignant transformation of cartilage tumours (Supplementary Fig. 7, Figure 4g, and Supplementary Fig. 9d). We also further characterized the patient specific clusters in Supplementary data – “*Characterization of High1, High2, and Cos cell clusters*”.

Corresponding updates: Line 305-331; Supplementary Fig. 7; Figure 4g; Supplementary Fig. 9d; Supplementary table 5; Supplementary data – “*Characterization of High1, High2, and Cos cell clusters*”; Supplementary Fig. 11

Comment 2: The inclusion of a chondroblastic osteosarcoma is not well explained and it can be discussed whether it makes sense since it is a different sarcoma type. It can be seen in figure 1A that it clusters separate. The advice would be to remove that sample

Response:

We included the chondroblastic osteosarcoma sample as a control to avoid over-integration when removing batch effect and patient-specific variances, given that it is a distinct cluster containing atypical neoplastic chondrocytes.

Corresponding updates: Line 111-112, 152-154

Comment 3: Immune cells are found in the scRNA-seq analysis, it is a pity that a more in depth analysis of the immune microenvironment is lacking, which would have improved the paper

Response:

We have further analysed the tumour-associated stromal cells, including immune cells, and included this section in the supplementary data and Supplementary Fig. 10-13, while our focus remains on the malignant transformation of cartilage tumours.

Corresponding updates: Supplementary data - "*Characterization of Prol, Stro, and Leuk cell clusters*"; Supplementary Fig. 11-13; Supplementary Table 1.9-1.10, 2.8-2.9

Comment 4: The CNV analysis per tumor could be extended. It would be nice to see the copy number variation per cell and see how heterogeneous they are within a patient. Comparison between the different samples is less informative as CNVs are known to be highly heterogeneous between samples.

Response:

We have added a heatmap showing the heterogeneity of inferred CNV of individual cells as Supplementary Fig 4a (bottom). We acknowledge the intra- and inter-tumoral heterogeneity of CNVs at single-cell resolution. Fortunately, we found a region at chromosome 6 commonly lost in low- and medium-grade tumours, indicating a distinct molecular subtype with shared molecular signatures. This is one of the reasons why we focused on these samples in this study.

Dropout event where a gene is not detected in all cells of the same cell type (cluster) is one of the limitations of current single-cell sequencing protocols. Advanced sequencing protocols and computational methods need to be developed to comprehensively analyse the CNV heterogeneity of individual cells. Given the limitation of current single-cell RNA sequencing technology, we analysed CNV at cluster level, which discovered the key CNV events during the progression of Chon1 and Chon2 cells. And we further analysis the copy number gain region specific to Chon2 cells in patient Low_2 (Supplementary Table 3.2; Supplementary Fig 4d).

Corresponding updates: Line 196-202; Supplementary Fig 4a, 4d; Supplementary Table 3.2

Comment 5: It is tested whether the cellular response to ER stress is specific to central chondrosarcomas (line 280). In order to test this only one peripheral chondrosarcoma sample was analyzed. A larger panel of tumors should be analyzed before conclusions can be drawn. Here the inclusion of an unrelated tumor such as osteosarcoma as a control can be considered.

Response:

We agree that a larger dataset is needed to draw a conclusion regarding ER stress in central chondrosarcomas. Chondrosarcoma is a rare tumour, and peripheral chondrosarcoma accounts for only 10% of cases. It is difficult to recruit enough patients to test this hypothesis in a short period of time in Hong Kong. We have extensively revised this section to validate that the Chon2 cell population is associated with conventional chondrosarcomas. To focus on conventional chondrosarcoma, we used a new conventional chondrosarcoma sample to replace the dedifferentiated chondrosarcoma (Supplementary Fig. 7).

Corresponding updates: Line 305-331; Supplementary Fig. 7

Comment 6: Clinical information and IDH /EXT status of the samples should be provided

Response:

We have retrieved the IDH1/2 status of the samples from clinical record. Coupled with IDH1/2 status determined by single-cell RNA sequencing, we summary the status of IDH1/2 in supplementary table 6. Due to the limitation of single-cell RNA sequencing, the read coverage is, unfortunately, too low to determine the status of EXT. Besides, EXT is not a routine test in our hospital. We are sorry that we cannot provide EXT status in this study. Other clinical information was summarized in Supplementary Fig. 1a.

Corresponding updates: Line 355-356; Supplementary Table 6

Comment 7: Reference 25 does not seem to be right? Reference 25 contain bulk RNA samples from hepatocellular carcinomas, not chondrogenic tumor samples?

Response:

We apologize for the incorrect reference and have corrected it (Integrated molecular characterization of chondrosarcoma reveals critical determinants of disease progression; 10.1038/s41467-019-12525-7).

Corresponding updates: Line 859-861

Comment 8: Reference 3: needs update to 2020 WHO classification 5th edition

Response:

We have updated the reference to the 2020 WHO classification 5th edition.

Corresponding updates: Line 794-795

Comment 9: It is not clear how the cell populations described in lines 285-289 were established and what the exact characteristics are.

Response:

We have provided gene markers and gene set enrichment analysis characterizing individual cell clusters in Supplementary Table 5 for readers who are interested in the second bath of samples.

Corresponding updates: Line 308-313; Supplementary Table 5

Comment 10: It is not clear what kind of primary tumour tissue was implanted subcutaneously to NOD SCID mice (line 321)

Response:

Given the slow growth of primary low- and medium-grade tumors, we employed a PDX-mouse model derived from a high-grade chondrosarcoma with IDH2 mutation (H172R) to test the impact of ER stress on chondrosarcoma growth.

Corresponding updates: Line 355-356; Supplementary Table 6

Comment 11: The survival curves shown merely represent the diagnosis / tumor type than a gene expression cluster: this was not shown to be independent and therefore these curves are misleading and should be removed

Response:

We understand and appreciate the concern that has been expressed regarding the potential for our curves to be misleading. To address this concern and provide additional clarity, we have included Supplementary Fig. 9a, which features survival curves of histological grades for comparison with our expression groups. Although both sets of survival curves are meaningful, the curves of our expression groups are more statistically significant. Furthermore, our expression groups have the ability to stratify patients with low- or medium-grade tumours into a group with a five-year survival rate of 100% and other groups with inferior survival rates.

Corresponding updates: Line 433; Supplementary Figure 9a

Comment 12: The immunohistochemistry shown only confirms the expression, not the differential expression between groups, and is therefore not informative. Moreover, the DDIT3 staining does not seem specific as it should be nuclear staining only.

Response:

We conducted additional experiment using 8 benign tumours and 13 chondrosarcomas. Differential expression between benign and malignant tumours was evaluated by immunoreactive scores (Fig. 4g, Supplementary Fig. 9d, and Supplementary Table 7). The immunoreactive score of malignant tumours was higher than benign lesion.

Corresponding updates: Line 452-457; Figure 4g; Supplementary Fig. 9d; Supplementary Table 7

Reviewers' comments:

Reviewer #1 (Remarks to the Author):

The authors addressed all comments. Accept to publish

Reviewer #2 (Remarks to the Author):

The authors have addressed all my questions.

Reviewer #3 (Remarks to the Author):

The authors have made some changes to the manuscript but did not address all of our comments, and concerns remain with regards to the design of the study, data analysis and interpretation, which prohibits publication of this manuscript in its current form.

Re comment 1:

The concern about the unfair comparison between the chondrosarcomas due to high heterogeneity still stands. The authors did not include more samples for single cell sequencing analysis or show that the results are not because of a patient comparison. Characterization of patient specific clusters is not really informative as it is not known if resulting high expressed genes reflect the patient or tumor biology. The addition of figure 4g helps support the ER claim. However, the grade of the malignant tumors in this figure is not mentioned. If the CHOP is also higher expressed in the high grade malignant tumors, then the ER stress should also be visible in the single cell data of the high grade chondrosarcomas. Of note, why is it that ER stress is important for malignant transformation, but not retained in high grade chondrosarcomas as seen in the single cell data?

Re comment 2:

With the response on comment 2 it sounds like the authors are forcing the clustering by adding the chondroblastic osteosarcoma sample. As the cell types are mostly separated by patients, it seems that the data is under-integrated and batch correction was not optimal. It is highly suggested to remove the chondroblastic osteosarcoma sample.

Re comment 5:

The claim that cellular response to ER stress is a marker for conventional chondrosarcoma is still not valid. Authors removed the focus on the peripheral chondrosarcomas in the text. Therefore it is not clear anymore why ER stress should be a marker for conventional chondrosarcoma. In order to address and validate this claim, a large panel consisting of multiple peripheral chondrosarcomas and conventional sarcomas should be investigated. Thereby could only be looked at the genes involved in the ER stress response. Generating new single cell sequencing data should not be necessary.

Re comment 6:

Single cell sequencing data is very sparse and shallow. This should not be the method to detect IDH mutations. IDH mutated samples could be misclassified as WT when the variant allele frequency is low of the IDH mutation / high contamination of normal cells.

Re comment 11:

This is not how showing independency works. The survival curves of the benign group and medium group in figure 9a are almost identical to the survival curves in 9c which includes also benign and

medium groups.

Re comment 12:

the authors do not comment on the lack of specific nuclear staining for DDIT3. Also, the amount of tumors used for validation is still too low to draw conclusions. Different locations and different histological grades should be represented when the authors want to show that DDIT3 is involved in malignant transformation

Response to referees

Reviewer #3 (Remarks to the Author):

The authors have made some changes to the manuscript but did not address all of our comments, and concerns remain with regards to the design of the study, data analysis and interpretation, which prohibits publication of this manuscript in its current form.

Re comment 1.1:

The concern about the unfair comparison between the chondrosarcomas due to high heterogeneity still stands. The authors did not include more samples for single cell sequencing analysis or show that the results are not because of a patient comparison. Characterization of patient specific clusters is not really informative as it is not known if resulting high expressed genes reflect the patient or tumor biology. The addition of figure 4g helps support the ER claim. However, the grade of the malignant tumors in this figure is not mentioned. If the CHOP is also higher expressed in the high grade malignant tumors, then the ER stress should also be visible in the single cell data of the high grade chondrosarcomas.

Response 1.1:

In Figure 4, we have used one widely used cell line (SW1353) which was derived from a medium-grade central chondrosarcoma, primary high-grade chondrosarcoma, and a PDX mouse model generated using the same high-grade tumour. We attempted to use cancer cells derived from a benign tumour to perform the PDX-mouse experiment, but it found to be too slow-growing to conduct functional analysis, which is similar to human cases (Ref. 2). Our single-cell data (Fig. 1d) and public bulk microarray data (Fig. 4f) both show that CHOP is downregulated also in high-grade tumours. Results from chondrosarcoma sample of different grade support our claim that ER stress affects tumour growth.

Re comment 1.2:

Of note, why is it that ER stress is important for malignant transformation, but not retained in high grade chondrosarcomas as seen in the single cell data?

Response 1.2:

Our hypothesis is that sustained ER stress has significant effects on immunosuppression, chemoresistance, angiogenesis, proliferation, and cancer cell survival (Ref 35-39). Prolonged and intense ER stress can lead to cell death via the CHOP-mediated apoptosis pathway (Ref 40). Our trajectory analysis revealed that ER stress levels increased and then decreased during the development of differentiated chondrosarcomas (Fig. 4c-g).

On the other hand, high-grade tumours might not exhibit the same progression pattern of ER stress as differentiated tumours and can occur *de novo* as the CNV loss at chromosome 6 observed in differentiated tumours was not detected in high-grade tumours (Supplementary Fig. 4a).

Re comment 2:

With the response on comment 2 it sounds like the authors are forcing the clustering by adding the chondroblastic osteosarcoma sample. As the cell types are mostly separated by patients, it seems that the data is under-integrated and batch correction was not optimal. It is highly suggested to remove the chondroblastic osteosarcoma sample.

Response 2:

To reassure reviewer 3, we have generated a new UMAP without chondroblastic osteosarcoma sample. The pattern of UMAP without chondroblastic osteosarcoma (Fig. R1 left) is consistent with the UMAP we presented in the manuscript (Fig. R1 medium). Fisher's exact test further showed that the cell types present in the manuscript (Chon1, Chon2, High1, High2, Pro, Stro, and Luek) are significantly correlated with their corresponding cell cluster (0,4,3,1, 5, 6, and 2; Fig. R1 right). Chondroblastic osteosarcoma serves as a good control to confirm the effectiveness of the integration analysis while did not affect the integration.

On the other hand, the quality of the integration was support by the evidence that proliferating cells (Prol), leukocytes (Leuk), and other stromal cells (Stro) from different patients were integrated into a distinct cell cluster, respectively. Neoplastic cells are known to be patient-specific, making the integration of cancer cells challenging. To further address your concerns regarding under-integration, we have analysed a new batch of samples individual without integration, confirming that Chon1 and Chon2 are commonly present in central chondrosarcoma (Fig. R2). Additionally, we re-analysed the first batch of samples using the same method and obtained consistent results (Fig. R3). This evidence demonstrates that the identification of Chon1 and Chon2 in the cell atlas is reproducible when analysing samples individually, indicating that our integration approach is effective.

Figure. R1 UMAP plots showing a comparison of the chondrosarcoma atlas without (left) or with (middle; Main Fig. 1a) the chondroblastic osteosarcoma sample. The heat map (right) shows odds ratios between cell cluster labels defined in the cell atlas with (rows) or without (column) the chondroblastic osteosarcoma sample.

Figure R2 (Supplementary Fig. 7) Chon2 cluster featured by response to ER stress is a marker for central chondrosarcoma

Figure R3. Chon2 and Chon1 scores of cell clusters identified in patients individually.

Re comment 5:

The claim that cellular response to ER stress is a marker for conventional chondrosarcoma is still not valid. Authors removed the focus on the peripheral chondrosarcomas in the text. Therefore it is not clear anymore why ER stress should be a marker for conventional chondrosarcoma. In order to address and validate this claim, a large panel consisting of multiple peripheral chondrosarcomas and conventional sarcomas should be investigated. Thereby could only be looked at the genes involved in the ER stress response. Generating new single cell sequencing data should not be necessary.

Response 5:

Indeed, all samples collected are conventional central chondrosarcomas (CCCS) which represents around 85% of chondrosarcomas. As a result, we have updated the title, abstract, and the main text to restrict our scope of study to CCCS only.

Re comment 6:

Single cell sequencing data is very sparse and shallow. This should not be the method to detect IDH mutations. IDH mutated samples could be misclassified as WT when the variant allele frequency is low of the IDH mutation / high contamination of normal cells.

Response 6:

We agree that the single cell sequencing data is very sparse and shallow, with potential false negative results for IDH mutation. However, this is not the central focus of our study, and we do not intend to have subgroup analysis between IDH-mutated and IDH wild-type samples. Therefore, we shall remove the results of IDH mutation from the manuscript.

Re comment 11:

This is not how showing independency works. The survival curves of the benign group and medium group in figure 9a are almost identical to the survival curves in 9c which includes also benign and medium groups.

Response 11:

We did not attempt to demonstrate independence by comparing the two survival curve plots. Tumour grades were determined by pathologists, while expression groups were generated based on the expression profile. These two factors are fundamentally different. By presenting the two survival curve plots, we aim to show an independent method (expression profile) for stratifying patients into groups with similar, if not better, prognostic value compared to histological grade.

Re comment 12.1:

The authors do not comment on the lack of specific nuclear staining for DDIT3

Response 12.1:

The latest batch of DDIT3 staining, which utilized citrate buffer for antigen retrieval, exhibited specific nuclear localization. We recognize that the previous batch of DDIT3 staining, which employed trypsin for antigen retrieval, lacked specific nuclear staining. The optimized antigen retrieval method contributed to the enhanced staining results, which is already included in our manuscript (Fig. R4).

Fig. R4 (Supplementary Fig. 9d) Representative images show DDIT3/CHOP immunohistochemistry staining with negative (0-1), mild (2-3), moderate (4-8), and strong (9-12) immunoreactive scores. Scale bar = 250 μ m.

Re comment 12.2:

Also, the amount of tumors used for validation is still too low to draw conclusions. Different locations and different histological grades should be represented when the authors want to show that DDIT3 is involved in malignant transformation

Response 12.2:

We agree that the number of cases used for validation is too low to draw conclusions. The results of DDIT3 stains in our small cohort of samples may suggest potential clinical application, with higher expression in chondrosarcomas compared to enchondroma. Further study involving a large number of chondrosarcomas in different locations and different histological grades, as well as other chondroid lesions and histological entities entering into the differential diagnosis, will be needed to comprehensively evaluate the diagnostic sensitivity and specificity of DDIT3 for clinical applications.

REVIEWERS' COMMENTS:

Reviewer #4 (Remarks to the Author):

The manuscript by Su and colleagues focuses on conventional chondrosarcomas where the authors identified the involvement of ER in malignant transformation by single-cell RNA sequencing. They also performed a PDX mouse model analysis in order to confirm their findings.

The authors replied to reviewers comments discussing and clarifying alle the raised issues, and modified the manuscript accordingly. In particular, they provided extensive and satisfactory point-by-point replies to reviewer 3 comments, significantly improving the manuscript which is now suitable for publication.